# T Cell Peptide Prediction, Immune Response, and Host–Pathogen Relationship in Vaccinated and Recovered from Mild COVID-19 Subjects

**DOI:** 10.3390/biom14101217

**Published:** 2024-09-26

**Authors:** Iole Macchia, Valentina La Sorsa, Alessandra Ciervo, Irene Ruspantini, Donatella Negri, Martina Borghi, Maria Laura De Angelis, Francesca Luciani, Antonio Martina, Silvia Taglieri, Valentina Durastanti, Maria Concetta Altavista, Francesca Urbani, Fabiola Mancini

**Affiliations:** 1Department of Oncology and Molecular Medicine, Istituto Superiore di Sanità, 00161 Rome, Italy; iole.macchia@iss.it (I.M.); marialaura.deangelis@iss.it (M.L.D.A.); silvia.taglieri@iss.it (S.T.); 2Research Promotion and Coordination Service, Istituto Superiore di Sanità, 00161 Rome, Italy; valentina.lasorsa@iss.it; 3Department of Infectious Diseases, Istituto Superiore di Sanità, 00161 Rome, Italy; alessandra.ciervo@iss.it (A.C.); donatella.negri@iss.it (D.N.); martina.borghi@iss.it (M.B.); fabiola.mancini@iss.it (F.M.); 4Core Facilities, Istituto Superiore di Sanità, 00161 Rome, Italy; irene.ruspantini@iss.it; 5National Center for the Control and Evaluation of Medicines, Istituto Superiore di Sanità, 00161 Rome, Italy; francesca.luciani@iss.it (F.L.); antonio.martina@iss.it (A.M.); 6Neurology Unit, San Filippo Neri Hospital, ASL RM1, 00135 Rome, Italy; valentina.durastanti@aslroma1.it (V.D.); mariac.altavista@aslroma1.it (M.C.A.)

**Keywords:** COVID-19, bioinformatics, ELISpot, T cell epitopes, T peptide, neutralizing antibodies

## Abstract

COVID-19 remains a significant threat, particularly to vulnerable populations. The emergence of new variants necessitates the development of treatments and vaccines that induce both humoral and cellular immunity. This study aimed to identify potentially immunogenic SARS-CoV-2 peptides and to explore the intricate host–pathogen interactions involving peripheral immune responses, memory profiles, and various demographic, clinical, and lifestyle factors. Using in silico and experimental methods, we identified several CD8-restricted SARS-CoV-2 peptides that are either poorly studied or have previously unreported immunogenicity: fifteen from the Spike and three each from non-structural proteins Nsp1-2-3-16. A Spike peptide, LA-9, demonstrated a 57% response rate in ELISpot assays using PBMCs from 14 HLA-A*02:01 positive, vaccinated, and mild-COVID-19 recovered subjects, indicating its potential for diagnostics, research, and multi-epitope vaccine platforms. We also found that younger individuals, with fewer vaccine doses and longer intervals since infection, showed lower anti-Spike (ELISA) and anti-Wuhan neutralizing antibodies (pseudovirus assay), higher naïve T cells, and lower central memory, effector memory, and CD4hiCD8low T cells (flow cytometry) compared to older subjects. In our cohort, a higher prevalence of Vδ2-γδ and DN T cells, and fewer naïve CD8 T cells, seemed to correlate with strong cellular and lower anti-NP antibody responses and to associate with Omicron infection, absence of confusional state, and habitual sporting activity.

## 1. Introduction

The coronavirus disease 2019 (COVID-19) pandemic has led to the rapid development and distribution of several vaccines, effectively reducing the spread of the virus, disease severity, hospitalizations, and deaths. However, the continuous emergence of new viral variants affecting the binding sites of neutralizing antibodies (nAbs) necessitates additional efforts to search for new treatments and vaccines [1,2,3,4,5]. Immune homeostasis appears to play a critical role in protecting against SARS-CoV-2 infection, although the specific mechanisms and host factors involved in disease evolution are not fully understood [6,7,8].

In general, SARS-CoV-2 natural infection can elicit both humoral and cellular immune responses [9,10,11,12] with a long-lasting protection provided by antigen-specific T cell immunity as compared to the antibody response [13,14,15,16]. Although neutralizing antibodies have been associated to protection in vaccinated individuals, no clear universal correlate of protective immunity has been validated and standardized so far in COVID-19 [17], due to the evasion from neutralization by new emerging variants [18]. However, the magnitude and functionality of T cell responses have been linked to disease severity and to the ability to mount an effective immune response, crucial for recognizing and eliminating virus-infected cells. Individuals with a strong and diversified T cell response may exhibit milder symptoms or even be asymptomatic, indicating a more effective control of the infection [19]. In particular, CD8 T cells have been shown to play a key role in mitigating disease severity, offering long-term immune protection against mild COVID-19 [20]. Memory T lymphocytes retain information about previously encountered pathogens, enabling a faster and more effective response upon re-exposure, thus conferring a stable response throughout convalescence [21]. Polyfunctional T lymphocytes, especially interferon-γ-secreting CD4 T cells, are known to be critical in this response [22,23].

A comprehensive SARS-CoV-2 vaccine should include both B cell epitopes for eliciting nAbs and T cell epitopes for robust and long-term immunity [24]. It has been demonstrated that some vaccines are able to induce strong T and B cell responses [25,26]. Recent literature strongly supports the use of updated vaccines in order to induce potent humoral and cellular immune responses simultaneously against all known variants of the SARS-CoV-2 virus [27]. Moreover, the induction of SARS-CoV-2 T cell immunity is a central goal for vaccine development, and of particular importance for patients with congenital or acquired B cell deficiencies [28]. Hence, assessment of cellular immune response may complement antibody testing to determine correlates of protection, especially in immunocompromised individuals [29].

Novel vaccine strategies include immunoinformatic/computation-based peptide vaccines, which have shown promise against viral pathogens, as they can be engineered with specific antigenic regions, reducing adverse reactions [30]. In this regard, in silico studies predicted numerous potentially immunogenic SARS-CoV-2 T cell epitopes, with good global coverage [31] and many MHC-class I CD8+ T cell epitopes have been identified so far. The in silico-based prediction methods may reduce the number of antigenic peptides for experimental testing, thus saving time and cost [32,33]. Generally, linear T cell epitopes offer more reliable results, compared to linear B cell or discontinuous epitopes [34]. However, discordance in the prediction of some of them highlights the need for further improvement, and more specialized tools [35]. As an added benefit, peptide-based vaccine candidates are cost-effective and safe [36].

For better and longer-lasting protection against COVID-19, next-generation multi-epitope vaccines [37], able to induce both humoral and cellular response, have been designed, based on peptide pools [38,39]; in some cases, these vaccine candidates have already reached the phase I clinical stage [28,40]. Another multi-epitope approach is represented by longer peptides that exhibit an improved stability and solubility compared to shorter peptides, thus reducing their risk of degradation and increasing their suitability for vaccine formulation and storage [41,42,43,44,45]. Further strategies, such as multiple allele epitope engineering, are expected to enhance the efficacy of peptide-based vaccines [46].

Based on these assumptions, a study has been designed aimed at the identification, by means of suitable computer prediction tools, of new potentially immunogenic HLA-A*02:01-restricted epitopes of SARS-CoV-2 proteins, specific for cytotoxic T lymphocytes. In addition, while vaccines licensed so far are based only on the Spike protein, both structural and non-structural proteins should be considered as potential vaccine targets. In fact, early recognition of non-structural proteins by the immune system may inhibit virus replication and spread [47]. Therefore, we considered epitopes derived not only from the Spike protein but also from some non-structural proteins that play a pivotal role in viral replication (Nsp1, Nsp2, Nsp3, and Nsp16).

Subsequently, we tested the immunogenicity of the selected peptides on cryopreserved peripheral blood mononuclear cells (cPBMCs) derived from healthy subjects’ blood who were all vaccinated against SARS-CoV-2 and recovered from paucisymptomatic/mild COVID-19. Several studies have utilized IFN-γ ELISpot, intracellular staining of cytokines (ICS), or non-cytokine activation-induced marker (AIM) by flow cytometry (FC) on PBMCs to characterize anti-SARS-CoV-2 T-lymphocyte responses [48]. Here, the ELISpot assay was selected as the more sensitive approach for our purposes [49]. The subjects enrolled in this study were also characterized for their virus-specific antibody response.

In addition, multiparametric flow cytometry (MFC) was employed to determine the circulating T cell subsets, including some less represented subpopulations, and their related naïve/memory status [50,51,52,53]. Among them, a small subpopulation of circulating CD4+CD8+double positive (DP) T cells has been described in healthy and pathological conditions, with unique and well-defined functions [54]. DP T cells exhibit memory-like features, with a predominant effector memory (EM) and central memory (CM) phenotype and can be distinguished in two distinct subsets (CD4dimCD8hi or DP1 and CD4hiCD8dim or DP2). These cells may represent a useful marker to predict the disease outcome, since they are significantly reduced in severe COVID-19 [55]. Other relevant subpopulations in the context of COVID-19 are represented by the CD4-CD8- double negative (DN) subset that possess both innate and adaptive immune functions, differing from conventional CD4+ and CD8+ T cells [56]. Despite their low frequencies, DN T cells, have a role in orchestrating immune responses through cytokine production and they display effector functions associated with pathology development [57]. Among DN, the gamma-delta (γδ) lymphocytes are a subset with a restricted receptor repertoire, sharing some characteristics with NK cells, as both are often associated with innate immunity. These cells, endowed with a cytolytic activity, are able to readily respond to a wide range of both infectious and non-infectious stressors [58,59,60]. In particular, γδ T cells expressing the Vδ2 TCR chain predominate in the peripheral blood and secondary lymphoid organs.

Various T cell perturbations have been described in COVID-19 patients with different degrees of severity [61,62,63,64,65].

Ultimately, we tried to define the complex interplay between all the gathered data, linking the humoral and cellular immune response to the host’s peripheral T cell memory profile, demographic characteristics, lifestyle, and the severity of the disease. We thus obtained some compelling evidence of the interrelationship among all the collected information, which could aid in tailoring therapeutic and vaccination interventions, and serve as a model in anticipation of a new pandemic.

## 2. Materials and Methods

### 2.1. Study Workflow

The study workflow can be visualized in Figure 1. Briefly, a preliminary in silico phase was conducted by bioinformatics tools, to identify HLA-A*0201-restricted epitopes of viral proteins, which had been poorly studied for their immunogenicity. After chemical synthesis, peptide immunogenicity was assessed using cPBMCs obtained from healthy donors who had been previously vaccinated and subsequently infected and recovered from mild COVID-19. Furthermore, ex vivo immune phenotyping for the determination of T cell memory status was conducted on fresh whole blood, while plasma was assessed to quantify total and neutralizing Abs directed against viral proteins. Subjects’ demographic, clinical, and lifestyle data were collected as well.

### 2.2. Selection and Synhtesis of Peptides

#### 2.2.1. T Cell Epitope Prediction

To predict potential immunogenic peptides, we used the amino acid sequences of the widely known Spike (YP_009724390.1) and non-structural protein (Nsp1—YP_009725297.1, Nsp2—YP_009725298.1, Nsp3—YP_009725299.1, and Nsp16—YP_009725311.1) accession numbers, collected in the database of the National Center for Biotechnological Information (NCBI) and derived from the Wuhan-Hu-1 (genome accession number NC_045512) reference isolate. These sequences were submitted to the NetMHCpan-4.1b EL algorithm available at https://services.healthtech.dtu.dk/services/NetMHCpan-4.1/ (accessed starting from 15 October 2020), which is based on eluted ligands (EL) data and, associating a value to each epitope, calculates a percentile Rank (% Rank) [66].

The following parameters were set:Input type: FASTAAllele selected: HLA-A*02:01Peptide Length: Any lengthOther fields: default values

The tool generated a list of peptides predicted to bind with high affinity to the HLA-A*02:01 allele, ranked in descending order of % Rank.

#### 2.2.2. Selection of 9-11mer Poorly Studied Ancestral Peptides

We then submitted the FASTA format sequence of each listed peptide with a % Rank below 1.1 to the IEDB Analysis Resource (https://www.iedb.org/result_v3.php?cookie_id=60c6fc&active_tab=Tcell%20Assays) (accessed starting from 15 October 2020) to exclude peptides that had already been shown to elicit a positive response in an IFN-γ ELISpot release assay by other authors.

The following parameters were set:Epitope: Linear peptideSequence: Exact match (inputting the sequence of each peptide)Assay: T cell, IFN-γ release ELISpotMHC Restriction: Class IHost: HumanOther fields: Default values

Peptides that had never been reported as positive in IFN-γ-ELISpot assays (whether tested or not) were selected for synthesis and subsequent immunological testing (by IFN-γ-ELISpot), as well as for further characterization as described below. The main scientific articles reporting immunogenicity studies of the selected peptides, extracted from the IEDB website after a query without a specified assay type, are also listed in Table 1.

#### 2.2.3. Selection of 9-11mer Mutant Peptides

Based on the reference genome accession numbers for Delta B.1.617.2 (MZ359841.1) and Omicron B.1.1.529 (BA.1) (OL672836.1), we selected the mutant Spike protein sequences of the Delta B.1.617.2 (QWK65230.1) and Omicron (BA.1) (UFO69279.1) virus strains and compared them to the ancestral Wuhan-1 strain protein (YP_009724390.1) using the free software Mega v11 (iGEM, Temple University, Philadelphia, PA, USA) (https://www.megasoftware.net/, accessed starting from 15 October 2020) [67]. The software created a new protein sequence alignment by matching the three FASTA format sequences using the ClustalW method. Upon examining the alignment, we identified four mutant peptides corresponding to four selected peptides from the Wuhan-1 strain (KA10w-KA10δ, KL9w-NL9o, VV11w-VV11oδ, and VV9w-VG9o).

#### 2.2.4. Long Peptides

Ultimately, our analysis also focused on two regions with a high rate of constant substitutions and deletions in the subunit S1 of the Wuhan-1 (YP_009724390.1) Spike protein sequences, belonging to Delta (B.1.617.2—QWK65230.1) and Omicron (B1.1.529 (BA.1)—UFO69279.1) VOCs, allowed us to select four long peptides (LP), approximately 30 aa long, including one pair beginning at position 135 and the other pair at 203 of the Wuhan-1 protein sequence. We also estimated the presence of strong binder epitopes (% Rank < 0.6) for the most representative HLA super-type, including HLA-A*0201, using the NetMHCpan-4.1b version, within these long peptide sequences (Appendix A).

#### 2.2.5. Control Peptides

A positive control (CEF) was designed by mixing in equal parts three HLA-A*02:01 restricted immune-dominant peptides of cytomegalovirus (CMV) NLVPMVATV (pp65), Epstein–Barr virus (EBV) CLGGLLTMV (LMP2), and influenza virus (FLU) GILGFVFTL (M1).

KIADYNYKL and YLQPRTFLL (two known immuno-stimulating Spike HLA-A*02:01 restricted peptides), as well as VF9 (an irrelevant peptide, VTWFHAIHF, to be used as negative control) were also selected for the synthesis.

#### 2.2.6. Selected 9-11mer Peptide Additional Characterization

NetCTLpan version 1.1: Available as a prediction method at the IEDB Analysis resource page (http://tools.iedb.org/netchop/) (accessed starting from 15 October 2020) [67], this tool integrates the prediction of peptide binding affinity to MHC class I molecules within the MHC class I antigen processing pathway. It combines the proteasomal cleavage score (C-score), which predicts the likelihood of protein cleavage at the C-terminus by the proteasome, with the TAP score, which indicates the transport efficiency by the transporter associated with antigen processing (TAP) proteins. For each selected peptide, the sequence in FASTA format was entered into the appropriate input field, the species was set to “human”, and the allele was specified as “HLA-A*02:01”, while all other parameters were kept at their default values. The output included a “% Rank”, which inversely correlates with the peptide's binding capacity to the MHC molecule of interest.SYFPEITHI: This is an online database (www.syfpeithi.de) (accessed starting from 15 October 2020) that uses an algorithm to assign a score to each amino acid at specific positions based on its frequency in natural ligands, T cell epitopes, or binding peptides [68]. Each peptide sequence in FASTA format was submitted to the appropriate input field (Epitope prediction), specifying the HLA-A*02:01 allele and the peptide length. The algorithm produced a score directly proportional to the binding affinity between the MHC molecules and their ligands.VaxiJen v2.0 algorithm: Available at https://ddg-pharmfac.net/vaxijen/VaxiJen/VaxiJen.html (accessed starting from 15 October 2020), this tool evaluates the probability of a given peptide being an antigen based on a trained model [69]. Each selected peptide sequence was submitted in the appropriate input field using the default antigenicity threshold setting of 0.4, and the target organism was set to “virus”. The output provided an antigenicity score for each peptide along with a qualitative prediction (probable antigen or non-probable antigen).

#### 2.2.7. Identity with Other Human Coronaviruses

The selected 9-11mer peptides were further evaluated according to the percentage of identity between SARS-CoV2 and other human coronaviruses (OC43, HKU1, NL63, and 229E). Single epitope sequences were subjected to analysis by means of 'NCBI Basic Local Alignment Search Tool (BLAST) (https://blast.ncbi.nlm.nih.gov/Blast.cgi) (accessed starting from 15 October 2020) and results are reported in Appendix A.

#### 2.2.8. Chemical Synthesis

Bio-Fab Research (Rome, Italy) synthesized the selected predicted SARS-CoV-2 9-11mer, the long peptides, and the control peptides, with a purity > 95%, in freeze-dried form. Peptides were resuspended in DMSO at a concentration of 40 mg/mL and used in culture at a final concentration of 10 μg/mL, avoiding repeated freeze–thaw cycles.

### 2.3. Subjects

Subjects were enrolled by ISS based on the following criteria:

#### 2.3.1. Inclusion Criteria

18–60 years of age.

Laboratory-confirmed SARS-CoV-2 infection and negativization (both determined by PCR or rapid antigen test).

Negativization occurred 30–90 days before the enrollment date.

Previous asymptomatic/mild COVID-19.

Good general health conditions.

Understanding and agreeing to comply with planned study procedures.

Written informed consent.

#### 2.3.2. Exclusion Criteria

Previous severe COVID-19 (pneumonia, hospitalization).

Concurrent metabolic diseases (obesity, diabetes, resistant hypertension, severe heart disease, tumors, rheumatic disease).

Chronic infectious disease (HIV, HBV, HCV).

Concomitant biological, antibiotic, immunosuppressive therapy.

Using of immuno-suppressive drugs during COVID-19.

Incapacity to understand the informed consent.

Withdrawal of the signed informed consent.

#### 2.3.3. Demographic, Clinical, and Lifestyle Aata Collection

On blood sampling day, subjects were interviewed to gather information relative to their SARS-CoV-2 vaccination and infection course as well as their past and current health conditions. The survey’s main data are included in Table 2 (subjects’ demographic/vaccination/infection main data), Appendix A (symptoms), and Appendix A (other clinical/lifestyle characteristics).

### 2.4. Blood Sampling

Blood draws were performed by specialized staff at S. Filippo Neri Hospital (ASL RM1). Briefly, 30 mL of venous blood were collected in lithium heparin Vacutainer tubes (Becton Dickinson, San Jose, CA, USA). Samples were processed within 2 h: an aliquot of fresh blood was committed to HLA testing and immune-phenotyping, and plasma was collected by centrifugation and immediately frozen at −80 °C, while PBMC were separated by Ficoll density gradient (Lymphoprep, Axis-Shield, Scotland, UK) and frozen in liquid nitrogen until the moment of use as already described [70].

### 2.5. Immune Cell Assays

#### 2.5.1. HLA Test

Positivity to HLA-A*02 was tested by staining 50 µL of fresh whole blood with a FITC-anti-human HLA-A*02 antibody (clone BB7.2, Biolegend, San Diego, CA, USA).

#### 2.5.2. ELISpot

After 3 rounds of plate washing with distilled sterile water, an anti-human IFN-γ antibody (clone 1, D1K, Mabtech, Nacka Strand, Sweden, EU) was added (10 µg/mL) for 18 h at + 4° C in 96-well nitrocellulose-bottomed plates (Merck-Millipore MSP4510, Burlington, MA, USA). Plates were then washed with DPBS (Corning, Corning, NY, USA) and incubated with DPBS supplemented with 10% FBS (Corning, NY, USA) for 2 h at 37 °C, to avoid non-specific staining.

Meanwhile, cPBMCs were thawed in 20% FBS-DPBS, in the presence of 20 µg/mL DNAse (Sigma, Livonia, MO, USA), centrifuged, washed with 10% FBS-DPBS and re-suspended in complete medium, composed of RPMI (Thermo Fisher Scientific, Gibco, Waltham, MA, USA), 10% FBS, Penicillin/Streptomycin, non-essential amino-acids, Na-pyruvate, HEPES (all form Lonza, Basel, Switzerland, EU), β-mercaptoethanol (Sigma, MO, USA), and a sub-optimal dose of DNAse (10 µg/mL).

After counting, 250,000 cPBMCs were left for 2 h at 37 °C in a controlled atmosphere of 5% CO_2_ and then seeded in duplicate wells for each condition, in a final volume of 200 µL of complete medium per well. Following the addition of the appropriate stimuli, the cells were cultured for 24 h at 37 °C in a controlled atmosphere of 5% CO_2_.

Selected-predicted peptides, as well as VF9 (as negative control), CEF, and PepTivator^®^ (SARS-CoV-2 Prot_S Complete and Prot_N, Miltenyi Biotec, Bergisch Gladbach, Germany, EU, as positive controls), were added at the final concentration of 10 µg/mL plus 1 µg/mL of anti-CD28 (BD Biosciences, San Jose, CA, USA), as a co-stimulation. As an additional positive control, staphylococcal enterotoxin B (SEB; Sigma-Aldrich, Munich, Germany, 2 µg/mL) was used, whereas medium only, anti-CD28 and DMSO (1:4000), were added as additional negative controls.

The development of the ELISpot assay was performed according to the Mabtech protocol: briefly, after culture, cells were removed, and the plate washed 5 times with DPBS. A biotinylated anti-IFN-γ antibody (clone 7-B6-1, Mabtech) was then added and cells were incubated for 2 h at room temperature. The plate was washed again 5 times with DPBS and a HRP enzyme-conjugated streptavidin (Mabtech) was added for 1 h at room temperature. After a new round of washing with DPBS (5 times), the TMB substrate (Mabtech) was added for about 20 min. The plate was finally washed with tap water and allowed to dry for at least 24 h, after which it was read with Aid iSpot instrumentation using AID ELISpot software 7.0 iSpot.

#### 2.5.3. Characterization of Peripheral Blood T Cell Naïve-Memory Status

Whole blood was stained with a panel constituted of a 7-color mixture of fluorochrome-conjugated Abs (Appendix A), based on DuraClone technology (Beckman Coulter, Life Science Europe, Geneva, Switzerland) consisting of 5 conjugated Abs (anti-CD3, -CD4, -CD8, -CD45RA, -CCR7) in dry formulation, integrated with 2 dropped-in Abs (anti-CD45 and anti-Vδ2) in liquid formulation. Dried reagents have already proven to yield high reproducibility and efficient standardization in large-scale projects such as the ONE study [71,72]. Based on expression of CD45RA and CCR7, we defined the following naïve/memory subsets within CD3+, CD4 single positive (sp), CD8sp, DP1, DP2, DN and Vδ2+ γδ T cells: naïve (N, CD45RA+CCR7+), central memory (CM, CD45RA−CCR7+), effector memory (EM, CD45RA−CCR7−), and terminally differentiated (TD, CD45RA+CCR7−) cells (panel gating strategy is illustrated in Appendix A).

##### Staining Procedure

For each panel, 200 μL of whole blood were stained. Briefly, samples were incubated with the corresponding antibody cocktail for 15 min at room temperature in the dark. The red blood cells were lysed by adding FACS lysing solution (BD Biosciences, San Jose, CA, USA) at room temperature for 10 min. After washing with DPBS, cells were fixed in Formaldehyde 0.8% and stored at 4 °C in the dark until the acquisition within the next 2 h. Before acquisition, an equal volume of DPBS was added to the samples.

##### Flow Cytometry Acquisition and Analysis

Data acquisition was performed using a Gallios cytometer (Beckman Coulter, Brea, CA, USA) and analyzed by Kaluza (v.1.3) software (Beckman Coulter, CA, USA).

### 2.6. Plasma Antibody Assays

#### 2.6.1. Commercial Anti-Spike and Anti-NP ELISA

Plasma samples were tested for the qualitative detection of anti-SARS-CoV-2 Abs using the “Elecsys^®^ Anti-SARS-CoV-2” test kit, an electrochemiluminescence immunoassay by Roche Diagnostics (Basel, Switzerland), on the cobas e411 instrument. The assay is a double-antigen sandwich using recombinant nucleocapsid protein (NP) for the detection of total Abs (IgA, IgM, and IgG) against SARS-CoV-2 (anti-SARS-CoV-2 NP Abs). Results are reported as numeric values in the form of a cut-off index (COI; signal sample/cutoff) as well as in the form of a qualitative “non-reactive” (COI < 1.0; negative) or “reactive” (COI ≥ 1.0; positive) result.

Plasma samples were further tested using the “Elecsys^®^ Anti-SARS-CoV-2 S” test kit, recently released by Roche Diagnostics for the quantitative detection of Abs against SARS-CoV-2 spike receptor binding domain (anti-SARS-CoV-2 S-RBD Abs). The total antibody content in the sample is expressed as U/mL, traceable to the Roche Diagnostics internal standard for anti-SARS-CoV-2 S. This standard consists of an equimolar mixture of two monoclonal Abs that bind Spike-1 RBD at two different epitopes; 1 nM of these Abs correspond to 20 U/mL of the Elecsys^®^ Anti-SARS-CoV-2 S assay. The cut-off is 0.8 U/mL, and the linear range is up to 250 U/mL. Samples with a concentration > 250 U/mL have been diluted up to 1:1000 in specimen diluent.

#### 2.6.2. Neutralization Assay

##### Production of Pseudovirus Pseudotyped with Spike Variants

Lentiviral vector (LV) delivering Luciferase pseudotyped with Spike (LV-Luc/Spike) were generated by transient transfection of 293T Lenti-X cells as previously described [73,74,75]. In brief, 293T Lenti-X cells were transfected with the lentiviral transfer vector plasmid pGAE-Luc expressing the luciferase coding sequence, the packaging plasmid pAd-SIV3+ and the pseudotyping plasmids expressing Spike from Wuhan-1, Alpha, Delta or Omicron (BA.1, BA.2 or BA.4/5) utilizing the JetPrime transfection kit (Polyplus Transfection, Illkirch, France). Forty-eight hours post-transfection, the supernatants containing the LV-Luc/Spike were collected, filtered with a 0.45 μm pore size filter (Millipore), and stored in 0.25 mL aliquots at −80 °C.

##### Pseudovirus Titration and Neutralization Assay

Preparations of LV-Luc/Spike were tittered in Vero E6 cells (Cercopithecus aethiops derived epithelial kidney, ATCC C1008), as described [73]. Briefly, cells were plated in 96-well plates (Viewplate, PerkinElmer) for 48 h with serial dilutions of LV-Luc/Spike preparations. Luciferase expression was measured with a Varioskan luminometer (Thermo Fisher) using the britelite plus Reporter Gene Assay System (PerkinElmer).

For the neutralization assay, plasma serial 2-fold dilutions starting from 1:80 were incubated at 37 °C for 30 min in 96-deep well plates (Resnova, Roma, Italy) in duplicate with the LV-Luc/Spike providing final 2 × 10^5^ relative light units (RLU)/well. The mixture was then added to Vero E6 cells seeded in a 96-well Isoplate (Perkin Elmer, Groningen, The Netherlands) at a density of 2.2 × 10^4^ cells/well. Cell-only and virus-only controls were included. After 48 h, luciferase expression was measured using the britelite plus Reporter Gene Assay System. RLU numbers were transformed into percentage neutralization values, and relative to virus-only controls. Results were expressed as the inhibitory concentration (ID) 50, which corresponds to the dilution of plasma providing 50% inhibition of the infection (corresponding to neutralization), compared to the virus-only control wells. ID50 was calculated with a linear interpolation method [73,75].

### 2.7. Statistical Analysis

A SPSS (IBM-SPSS V25, IBM Corporate, New York, NY, USA) database collected all parameters under study (demographic, clinical, and lifestyle, as well as T and B immune response variables), which are shown in Appendix A, as well as the MFC variables listed in Appendix A. Some variables were the result of a calculation based on the collected data, such as the time interval between the 1st positive swab and sampling (PS-DT), the last vaccine dose and sampling (VS-DT), the last vaccine dose and 1st positive swab (VP-DT), and the 1st positive swab and the negative swab (PN-DT), all expressed in days (Appendix A). Outliers were appropriately eliminated.

Regarding the statistical analysis of ELISpot results, we compared absolute spot-forming cell counts (SFC) of each stimulus vs. VF-9 negative control as well as of mutant (o/δ) vs. Wuhan-1 peptides by non-parametric paired samples Wilcoxon test (Figure 2 in Results Section 3.3). VF9 irrelevant peptide always showed a less or comparable number of spot counts than medium, anti-CD28, and DMSO-only negative controls. Furthermore, due to the low average number of spot counts, a qualitative measure of peptide positivity was defined when the average of the spots produced was at least twice the average of the spots counted in the negative control condition (2 × VF-9 peptide average spot counts = cut-off value) (Table 3).

A non-parametric Spearman’s Rho test was applied to find a correlation between pairs of continuous variables (Table 1 and elsewhere in the text). The non-parametric Wilcoxon test was employed for correlated sample comparisons (Figure 2), while the Mann–Whitney U test was used for group-to-group comparisons (in the text).

#### Multiple Correspondence and Principal Component Analysis (MCA and PCA)

The R-package ‘Factominer’ (v.2.8) was used [76] for both MCA and PCA.

For MCA, the dataset consisted of 12 profiles of 18 features: 11 were the active variables (symptoms: Fever, Anosmia, Ageusia, Cough, Headache, Sore throat, Rhinorrhea, Muscle pain, Joint pain, Malaise fatigue, Confusional state), 3 were quantitative (Age, total symptoms, time-interval between 1st positive swab and sampling, this last hereafter defined as PS-ΔT), and 4 qualitative (VOC, Paracetamol, Intensive Sport Activity-Sport_Y/N, COVID-19 vaccine dose number-Vax dose#) supplementary variables. Rare symptoms (frequency < 3) were excluded, thus yielding a total of 11 symptoms included in the analyses.

For PCA the dataset consisted of 12 profiles of 67 features: 38 were active variables (cell subpopulations determined by flow cytometry), 14 were quantitative (immune responses: Spike LA-9, Spike KL-9w, 33-peptide response rate, Peptivator N, Peptivator S, anti-Spike Abs, anti-Wuhan nAbs, anti-NP Abs; context-related: time intervals between last vaccine dose and sampling (VS-ΔT), last vaccine dose and 1st positive swab (VP-ΔT), 1st positive swab and 1st negative swab (PN-ΔT) and PS-ΔT; personal: age, total symptoms), and 15 were qualitative (11 symptoms, VOC, intensive sport activity, paracetamol, vax dose#) supplementary variables.

The association between synthetic variables and supplementary variables was evaluated in terms of Pearson correlation for continuous variables and R2 for categorical ones.

## 3. Results

### 3.1. Peptide Selection

#### 3.1.1. 9-11mer Peptides In Silico Prediction

When the project started, several in silico prediction studies had already been conducted on the peptide's MHC-I receptor affinity, and the most promising peptides had already been tested for their immunogenicity by several research groups. At that point, we wondered about the immunogenic properties of the minor (less described in the literature) peptides. Therefore, we decided to concentrate on peptides that did not rank among the top best, although they still exhibited interesting % Rank values. We also sought to comprehend how mutations present in the most prevalent VOCs at that time could influence the cellular response toward individual peptides.

To this end, we obtained a set of representative CD8 T cell HLA-A*02:01-restricted peptides belonging to the Spike and to the Nsp1, Nsp2, Nsp3, and Nsp16 derived from the SARS-CoV-2-isolate Wuhan-Hu-1 protein sequences. To generate the peptide set, protein sequences were subjected to the NetMHCpan-4.1b algorithm, which assigns a % Rank inversely proportional to the peptides’ ability to bind to class I MHC [66]. Afterwards, a peptide selection based on the quality of the % Rank was performed, including peptides not yet tested for their immunogenicity in the current literature, to achieve 11 ancestral 9-11mer peptides derived from Spike (TL-9, LA-9, GL-9, VV-11w, KV-10, FV-10, VI-9, KA-10w, TL-10, VA-11, VV-9w), with a % Rank ranging from 0.041 to 1.043. We also included two already known and well-characterized immune-dominant peptides, namely KL-9w and YL-9, with % Ranks of 0.067 and 0.013, respectively. Three potentially immunogenic epitopes were selected as well, for each non-structural protein: TV-9, QV-9, VL-9 (Nsp1), TI-9, RT-9, FV-9 (Nsp2), IV-9, FL-10, KL-10 (Nsp3), and SL-10, WV-9, QL-9 (Nsp16), whose % Ranks ranged between 0.051 and 0.538, as shown in Table 1.

#### 3.1.2. 9-11mer VOC Mutated Peptide Pairs

Through comparison and mutation analysis performed by Mega software [77] based on the reference Wuhan-1 Spike protein sequence versus the principal VOCs circulating in Italy between June 2021 and May 2022 (Delta and Omicron BA.1, according to the COVID-19 Data Portal website) [78], four mutant peptides were selected: VG-9o and NL-9o both belonging to Omicron BA.1, KA-10δ present in Delta, and VV-11o/δ existing in both variants, pairing four previously selected Wuhan-1 peptides (VV-9, KL-9w, KA-10, and VV-11).

Of note, peptides whose % Rank is <0.5 are commonly defined as strong binders, while % Ranks ranging from 0.5 to 2 define weak binder peptides [66]. As we were interested in comparing paired ancestral-mutated peptides and in verifying correlation linearity between immunogenicity and % Rank, the current study included twenty strong binder peptides (% Rank < 0.5) and five peptides with a % Rank between 0.5 and 0.55, and some weak/not binders, such as the paired VV-9/VG-9o peptides (1.043 and 28,630 % Rank, respectively), the Delta mutant KA-10δ (1.841 % Rank), and the VA-11 peptide (0.735 % Rank). Mutant peptides’ % Rank was generally worse than that of their ancestral ones, except for Spike VV-11δ/o peptide whose % Rank resulted slightly better than the ancestral VV-11 one. Mutated peptides are coupled with the relative wild counterparts in Table 1, while details on the literature-based immunogenicity of the selected peptide pairs are described in the Appendix A [79,80,81,82,83,84,85].

**Table 1 biomolecules-14-01217-t001:** 9-11mer peptides.

	High Affinity					Algorithm for Peptide Selection	Other Algorithms		

	Low Affinity								
	Peptide ID	1st Aa Position	VOC °	Sequence	Already Tested by ^£^	EL NetMHCpan 4.1b % Rank	NetCTLpan % Rank	SYFPEITHI Score	VaxiJen Decision
Spike	KA-10w	947	W	KLQDVVNQNA		0.511	2.00	16	Antigen
	KA-10δ	947	D	KLQNVVNQNA		1.841 ^$^	5.00 ^$^	15	Non-antigen
	KL-9w ^#^	417	W	KIADYNYKL	Various assays [27,86,87,88,89,90,91,92,93,94,95,96,97,98]	0.067	0.30	26	Antigen
	NL-9o	417	O	NIADYNYKL	ICS [86], Multimer_staining [99]	0.352	1.00	25	Antigen
	VV-11w	610	W	VLYQDVNCTEV		0.290	0.30	nd	Antigen
	VV-11δ/o	610	D/O	VLYQGVNCTEV		0.237	0.30	nd	Antigen
	VV-9w	62	W	VTWFHAIHV	Multimer staining [99,100,101,102]	1.043 ^$^	2.00	15	Antigen
	VG-9o	62	O	VTWFHVISG		28630 ^$^	50.00 ^$^	11	Antigen
	FV-10	515	W	FELLHAPATV	Multimer staining [89]	0.377	7.00 ^$^	17	Antigen
	GL-9	857	W	GLTVLPPLL	Biol_activity [103], multimer staining [102,104], ELISpot [91]	0.259	2.00	22	Antigen
	KV-10	386	W	KLNDLCFTNV	ICS [96], multimer staining [97,104], ELISpot [92,105,106]	0.354	0.15	23	Antigen
	LA-9	821	W	LLFNKVTLA	ELISA [107], ELISpot [91,92,93], HTMA [108], ICS [96,109,110], c ytotoxicity [107,111], multimer staining [87,88,98,102,107,111]	0.105	0.80	22	Antigen
	TL-10	1136	W	TVYDPLQPEL		0.518	2.00	19	Antigen
	TL-9	109	W	TLDSKTQSL	Biol_activity [97], HTMA [108], ELISpot [91,92,94,112], multimer_staining [89,102,113,114]	0.041	2.00	25	Antigen
	VA-11	83	W	VLPFNDGVYFA	Multimer staining [89]	0.735	3.00	nd	Antigen
	VI-9	915	W	VLYENQKLI	ELISpot [91,115], multimer staining [102]	0.397	3.00	21	Antigen
	YL-9 ^#^	269	W	YLQPRTFLL	Various assays [27,86,87,88,89,90,91,92,93,94,95,96,97,98] and many others	0.013	0.05	26	Antigen
Nsp1	QV-9	15	W	QLSLPVLQV	ICS [116,117]	0.214	3.00	26	Antigen
	TV-9	103	W	TLGVLVPHV	ICS [111,117], multimer staining [90,98,102]	0.162	1.00	26	Antigen
	VL-9	38	W	VLSEARQHL	Biol_acitivity [116], ICS [116], multimer staining [89]	0.311	2.00	23	Antigen
Nsp2	FV-9	461	W	FLRDGWEIV	Biol_acitivity [116], ICS [116], multimer_staining [102]	0.360	0.80	24	Non-antigen
	RT-9	399	W	RLIDAMMFT	multimer staining [89,98]	0.438	1.50	17	Non-antigen
	TI-9	34	W	TLSEQLDFI	ICS [116],multimer_staining [102]	0.288	0.80	24	Antigen
Nsp3	FI-10	430	W	FLTENLLLYI	Multimer staining [89,107]	0.542	0.15	25	Non-antigen
	IV-9	1514	W	ILFTRFFYV	Biol_activity [103], ICS [109,111,112,113,114,115,116,117,118,119], multimer_staining [102,114]	0.051	0.01	23	Non-antigen
	KL-10	1407	W	KLINIIIWFL	Granzyme_B [94], Multimer staining [102]	0.538	0.05	27	Non-antigen
Nsp16	QL-9	266	W	QINDMILSL	Multimer staining [102]	0.313	2.00	27	Antigen
	SL-10	243	W	SLFDMSKFPL		0.435	0.15	24	Non-antigen
	WV-9	88	W	WLPTGTLLV	Biol_activity [86], multimer staining [102]	0.501	0.80	25	Antigen
		*p* by Spearman test vs. NetMHCpan					0.051	0.008	

° W: Wuhan-1, O: Omicron BA.1, D: Delta; ^£^ References are extracted from the iedb.org website; ^#^ known immuno-dominant peptide; ^$^ Outlier; nd: no peptide prediction matrices for “HLA-A*02:01” defined.

#### 3.1.3. 9-11mer Peptide In Silico Characterization

A deeper characterization of all selected peptides was performed by means of other specialized algorithms such as NetCTLpan-1.1 cell epitope prediction tool from the Immune Epitope Database and Analysis Resource (IEDB) [67] that combines proteasome cleavage, TAP transport, and MHC class I score, assigning a combined score and a specific % Rank (Table 1).

An additional score was also calculated by the SYFPEITHI algorithm (Table 1), which makes predictions based on natural ligands and epitopes of known T cells present in its database. Commonly, values above 20 are considered predictive of good binding. Most of the selected epitopes showed good scores (>20); no one had a score < 15, except for VG9o which showed a bad score (=11) [120].

Eventually, the VaxiJen assignment of the probability to be antigenic for each peptide was detailed in Table 1 [69]. Here, the virus was selected as the target organism, and the antigenicity threshold was set at 0.4. All epitopes were defined as suitable antigens, apart from Spike KA-10δ, Nsp2 FV-9 and RT-9, all Nsp3, and the Nsp16 SL-10 peptides.

#### 3.1.4. 9-11mer Peptides In Silico Prediction Scores: Correlation among Different Algorithms

Spearman's test revealed a nearly significant correlation between NetMHCpan-4.1b % Rank and NetCTLpan-1.1 % Rank (*p* = 0.051), while the SYFPEITHI score showed a stronger association with NetMHCpan-4.1b % Rank (*p* = 0.008) (Appendix A). At the same time, even though VaxiJen decision did not significantly group peptides based on their NetMHCpan-4.1b % Rank (*p* = 0.123) (Appendix A), all the most probable VaxiJen-defined non-antigen peptides were included in higher NetMHCpan-4.1b % Rank (>0.355), apart from Nsp3 IV-9 (0.051 % Rank) (Table 1).

#### 3.1.5. Forecasting 9-11mer Peptide Immunogenicity

Overall, based on NetMHCpan-4.1b % Rank and VaxiJen definition, we could speculate that the most promising epitopes in terms of immunogenicity could be those “probable antigen” peptides with a % Rank < 0.6. According to this definition, at the top of the classification we could find YL-9 (0.013 % Rank), TL-9 (0.041 % Rank), KL-9w (0.067 % Rank) and LA-9 (0.105 % Rank) peptides, all derived from the Spike protein. This prediction would have been partially verified once our investigation was carried out. In fact, only KL-9w and LA-9 peptides showed a good degree of immunogenicity post hoc.

In brief, Table 1 collects the complete sequence of each peptide, its first amino acid position in the Wuhan-1 strain protein sequence, the % Rank calculated by Net MHCpan-4.1b as well as the other algorithm scores. When available, we also reported the appropriate literature reference, and the type of test or analysis they were submitted to, as described in Jin et al. [42].

#### 3.1.6. Long Peptide Selection

Furthermore, we wanted to explore the possibility of assaying certain longer protein segments that were particularly rich in mutations by switching from the Wuhan-1 strain to the *Delta* and *Omicron* VOC strains. For this purpose, sequences of two long peptides spanning two regions potentially relevant for their mutational burden in Delta and Omicron BA.1 VOCs, both located in the S1 subunit, were investigated to explore their possible immunogenicity. The regions of interest were labeled as 135w and 203w according to their starting residue position in the Wuhan-1 protein sequence, and they were 32 and 28 amino acid long, respectively (135w amino acid sequence: FCNDPFLGVYYHKNNKSWMESEFRVYSSANNC, 203w: IYSKHTPINLVRDLPQGFSALEPLVDLP). These regions enclose several mutations from the ancestral strain, including amino acid substitutions, deletion, or insertion, as depicted in Appendix A. On this basis, we identified a 135δ and a 203o LP, which encompassed the Delta and the Omicron BA.1 mutations as compared to the Wuhan-1 sequence, both 30 amino acids long (respective sequence: FCNDPFLDVYYHKNNKSWMESGVYSSANNC and IYSKHTPIIVREPEDLPQGFSALEPLVDLP). By the way, we should point out that the 135w LP is conserved in Beta VOC, while 203w LP is conserved in Alpha, Gamma, and Delta VOCs.

Furthermore, the binding affinity of the potentially immunogenic T cell epitopes, enclosed within the selected LPs, was calculated by NetMHCpan-4.1b algorithm for the most represented HLA super-types, including HLA-A*02:01, as shown in Appendix A, along with other literature-based details of the selected LPs (Appendix A) [121,122,123,124]. None of the four LPs, Wuhan-1 and mutants, generated strong binder epitopes belonging to the HLA-A*0201 haplotype. Differently, they all showed, within their sequences, a strong binder epitope towards HLA-A*24:02 haplotype, which is a relatively rare allele in the population (incidence = 0–0.5%).

#### 3.1.7. 9-11mer and LP Peptide Sequence Identity with Other Coronaviruses

Selected (9-11mer and long) peptide sequences were compared for identity, and therefore for their putative cross-reactivity [125], towards other human Coronaviruses (OC43, HKU1, NL63 and 229E), using the NCBI Blastp platform (Appendix A). Most of the selected peptides exhibited a Sequence Identity (SI) of ≤70% with these human Coronaviruses, except for:

Spike GL-9, 77% SI with OC43;

Spike KV-10, 80% SI with NL63 and 229E;

Spike LA-9, 77% SI with HKU1;

Spike VI-9, 77% SI with OC43 and HKU1;

Nsp1 VL-9, 77% SI with NL63 and 229E;

Nsp16 QL-9, 77% SI with OC43 and 229E;

Nsp16 SL-10, 90% SI with OC43 and HKU1;

Nsp16 WV-9, 77% SI with OC43.

All selected long peptides showed an identity percentage lower than 50% compared to other coronaviruses. Based on these observations, we might speculate that immunological cross-reactions with other human Coronaviruses were unlikely, except for the SL-10 epitope of Nsp16, showing 90% SI with OC43 and HKUI viruses.

### 3.2. Subjects’ Characteristics

Twenty healthy subjects were enrolled between November 2021 and June 2022, being infected from July 2021 to May 2022. Among them, 14 out of 20 resulted positive for HLA-A*02 and were therefore included in our analysis. Table 2 reports the main characteristics related to the demographic features (sex at birth—hereafter referred as sex, for brevity—and age), the vaccine manufacturer (Johnson & Johnson, New Brunswick, NJ, USA; Pfizer, New York, NY, USA; Moderna, Cambridge, MA, USA), and the dose number, as well as the 1st positive swab date, the probable infecting SARS-CoV-2 VOC, the total symptom number, and the PS-, VS-, VP-, and PN-ΔT time intervals. In particular, the cohort comprised 11 female and 3 male subjects, with an average age of 51 years (min 18, max 58). According to their infection onset date (first positive swab), the potential VOC was estimated by referring to the COVID-19 Data Portal (CDP; https://www.covid19dataportal.org/) (accessed starting from 15 October 2020), an open-access data sharing [78]. On this basis, 5 subjects were presumably infected by Delta, 7 by Omicron BA.1, and 2 by Omicron BA.2 strain; all subjects showed a previous paucisymptomatic/mild COVID-19 course (median symptoms = 5, min 1–max 13). All subjects received one (14%), two (43%), or three (43%) anti-SARS-CoV-2 vaccination doses before the infection, while vaccine manufacturers were Johnson & Johnson (14%), Moderna (36%) and Pfizer (50%). A median of 70 days was determined in PS-ΔT (min 44, max 173), of 188 days (min 74, max 434) in VS-ΔT, of 107 days (min 4, max 261) in VP-ΔT, and of 12 days (min 7, max 26) in PN-ΔT.

**Table 2 biomolecules-14-01217-t002:** Subjects’ demographic/vaccination/infection main data.

Subj # ID	1st Positive Swab Date (d-m-y)	Sex	Probable VOC	Vaccine Dose Number	Vaccine Manufacturer *	Age (y)	Total Symptoms	PS-DT ^$^ (d)	VS-DT ^$^ (d)	VP-DT ^$^ (d)	PN-DT ^$^ (d)	
01	26.07.2021	F	Delta	1	M	31	1	116	147	31	9	
02	11.09.2021	F	Delta	2	J	51	7	69	170	101	10	
03	19.10.2021	F	Delta	2	J	55	13	52	206	154	26	
04	23.07.2021	M	Delta	1	P	18	4	165	169	4	15	
05	11.01.2022	F	Omicron BA.1	2	M	29	3	59	206	147	10	
06	31.12.2021	F	Omicron BA.1	3	P	58	2	70	89	19	11	
07	11.01.2022	F	Omicron BA.1	2	P	54	7	62	264	202	13	
08	07.01.2022	F	Omicron BA.1	3	P	51	3	66	74	8	10	
09	05.01.2022	M	Omicron BA.1	2	P	49	4	93	281	188	7	
10	03.01.2022	F	Omicron BA.1	3	P	51	5	108	129	21	16	
11	05.11.2021	M	Delta	2	M	41	5	173	434	261	16	
12	02.05.2022	F	Omicron BA.2	3	P	52	7	44	156	112	7	
13	21.04.2022	F	Omicron BA.2	3	M	53	5	55	226	171	13	
14	23.03.2022	F	Omicron BA.1	3	M	50	7	112	211	99	16	
	Frequency (n out of 14 Subjects)	M = 3	B 1.617.2 Delta = 5	1 = 2	J = 2	51	5	70	188	107	12	median
		F = 11	B.1.1.529 Om BA.1 = 7	2 = 6	M = 5	18	1	44	74	4	7	min
			B.1.1.529 Om BA.2 = 2	3 = 6	P = 7	58	13	173	434	261	26	max

* M = Moderna; J = Johnson & Johnson; P = Pfizer. **^$^** Time interval (DT) between 1st positive swab and sampling (PS), last vaccine dose and sampling (VS), last vaccine dose and 1st positive swab (VP), 1st positive swab and the negative swab (PN).

### 3.3. Assessment of T Cell Immune Responses to Selected Peptides

Immunogenicity analysis of the in silico predicted/selected peptides was performed on cPBMCs derived from 14 HLA-A*02: 01 positive healthy subjects’ blood samples by means of IFN-γ ELISpot assay and is reported in Figure 2. Most of the selected peptides exhibited a low and variable response across subjects. Comparing each peptide to the negative control, only one peptide poorly studied by functional ELISpot test so far, named Spike LA-9, exhibited a significant increase (*p* = 0.017) in terms of SFCs. The previously investigated KL-9w peptide confirmed its immunogenicity (*p* = 0.002) and the same was true for Peptivator S and Peptivator N (*p* < 0.001 and *p* = 0.001, respectively, after Bonferroni correction). The YL-9 peptide regarded as immunogenic showed a trend of increase in SFCs as compared to negative control, although not significant. Univariate analysis by Spearman test indicated that cellular response to the various immunodominant peptides and two viral protein peptide pools significantly correlated with each other (i.e., 33-peptide response rate with KL-9w and Peptivator S, as well as Peptivator S with Peptivator N, rho = 0.730, 0.621, 0.533 and *p* = 0.003, 0.018, 0.050, respectively).

We also investigated the response to LA-9 by ELISpot assay on cPBMCs derived from five non-HLA-A*02:01 vaccinated and recovered subjects, with characteristics comparable to the HLA-A*02:01 subjects described so far. A positive response was obtained in three out of five of these subjects (Appendix A), whose exact haplotype was not known, as we did not have the opportunity to perform a complete HLA-typing. We therefore queried the netMHCpan 4.1b algorithm to obtain predictions of MHC class I affinity for the most common alleles, as listed by the IEDB website, and observed that LA-9 appeared to be a strong binder candidate for two alleles (A*02:06 and B*08:01, % Rank 0.19 and 0.08, respectively) and a weak binder for many other alleles (with a % Rank ranging from 0.56 to 1.6) (Appendix A). Some of these alleles are described as relatively common (such as B*08:01, with a frequency of 6%).

**Figure 2 biomolecules-14-01217-f002:**
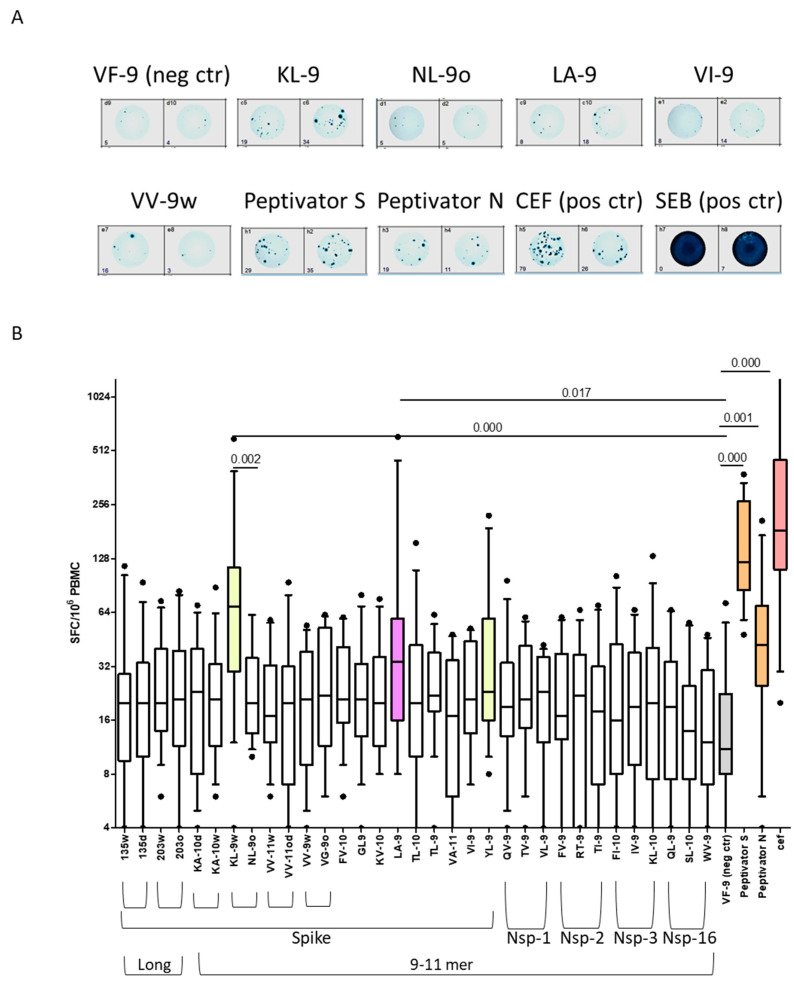
PBMC response to in silico selected peptides. (**A**) One representative experiment of a 24 h ELISpot assay on cryopreserved PBMCs derived from an HLA-A*02:01+ recovered and vaccinated subject (Subj#9). Here, we show the raw ELISpot data image of negative (VF-9) and positive (pool S, pool N, CEF, SEB) controls as well as Spike and Nsp in silico selected peptides, whose ratio vs. VF9 negative control resulted ≥2. (**B**) Boxplot (median and 10–90 percentile range) graph showing cumulative results of ELISpot assay on 14 subject cryopreserved PBMCs stimulated with in silico selected peptides as well as negative and positive controls. KL-9w and YL-9, in yellow, are already described as immunogenic peptides; LA-9, in purple, was the only poorly studied peptide to differ significantly from the VF-9 irrelevant peptide (negative control, in grey). Peptivator pool, in orange, and CEF, in pink, positive controls significantly differed from the negative control, as well. *p*-value = non-parametric Mann–Whitney test for paired samples, Bonferroni corrected. Black dots represent the outlier values.

#### 3.3.1. Qualitative Analysis of T Cell Response to Selected Peptides

A qualitative analysis of functional immune response has been carried out by estimation of the positive/negative value for each peptide tested on each subject. An SFC value that was at least double the SFC value of the negative control was defined as positive (Table 3). Hence, the immunogenicity rate for each peptide has been calculated as the ratio between the number of positive events and the number of tested subjects (n = 14). This rate varied between a minimum value of 0.07 and a maximum value of 0.93. In particular, the peptide LA-9 rate was equal to 0.57, being positive in more than half of the tested subjects. The immunodominant peptide KL-9w exhibited a rate of 0.79, while the above mentioned YL-9 showed a rate of 0.64. We could not define YL-9 peptide as immunodominant in our cohort of subjects, due to its lack of quantitatively significant responsiveness. However, this peptide has been described as immunogenic by several other authors [89] and we believe that the lack of observed response in our study could be attributed to natural biological variability, influenced by the small size of our sample group. The higher immunogenicity rate was attributed to 135δ long peptide (0.93) (Table 3), although weak and not significantly different from the negative control, when globally and quantitatively analyzed (Figure 2). All subjects’ cPBMCs showed a positive response to the CEF positive control and to the Peptivator S peptide pool (in this last case, except one), while four out of fourteen subjects’ cPBMCs did not react to the Peptivator N peptide pool. Only one subject did not respond to the Peptivator S, noteworthy for having experienced a greater number of symptoms and a longer infection time, despite having already received two doses of the vaccine. The response to the Peptivator N was more variable, with four unresponsive subjects.

Overall, following our ELISpot tests, we collected a consistent amount of information regarding the cellular response to the Spike and the Nuclear proteins, as well as the two immunodominant peptides in our subject cohort (KL-9w and LA-9). Additionally, we were able to assess the response rate across the 33 analyzed peptides (9-11mer and long), by calculating for each subject the incidence of the positive response (33-peptide response rate: median 0.29, min 0.06, max 0.73) (Table 3). Noteworthy, this response rate could reflect a comprehensive cellular response inclusive of both Spike protein and the four Nsp proteins under examination. 

**Table 3 biomolecules-14-01217-t003:** Qualitative evaluation of the cellular immune response: single peptide immunogenicity rate and determination of subjects’ response rate to the 33 peptides.

		Peptide	Subj#1	Subj#2	Subj#3	Subj#4	Subj#5	Subj#6	Subj#7	Subj#8	Subj#9	Subj#10	Subj#11	Subj#12	Subj#13	Subj#14	Immunogenicity Rate
Long	Spike	135W	+	−	−	+	+	−	+	+	−	−	−	−	−	−	0.36
		135δ	+	+	−	+	+	+	+	+	+	+	+	+	+	+	0.93
		203w	+	−	−	+	−	−	+	−	−	+	−	−	−	+	0.36
		203o	−	−	−	+	+	+	+	+	−	−	−	−	−	+	0.43
9-11 mer		KA-10w	−	−	−	+	+	−	+	−	−	−	−	−	−	+	0.29
		KA-10δ	−	−	−	+	+	−	+	−	−	−	−	−	−	+	0.29
		KL-9w	+	−	+	+	+	+	+	+	+	+	−	+	−	+	0.79
		NL-9o	+	−	−	−	−	+	+	−	−	−	+	−	−	+	0.36
		VV-11w	−	−	−	−	−	+	+	−	−	−	−	−	−	+	0.21
		VV-11oδ	−	−	+	−	+	−	−	−	−	−	−	−	−	−	0.14
		VV-9w	+	−	−	−	+	+	+	−	+	−	−	−	−	−	0.36
		VG-9o	−	−	−	+	−	+	+	−	−	−	−	+	−	+	0.36
		FV-10	+	−	−	+	+	+	+	−	−	−	−	+	−	+	0.50
		GL-9	+	−	+	+	−	−	+	−	−	−	−	−	−	+	0.36
		KV-10	−	−	−	−	+	+	+	−	−	−	−	−	−	−	0.21
		LA-9	−	−	+	+	+	−	+	+	+	−	−	−	+	+	0.57
		TL-10	−	−	−	+	+	−	+	+	−	−	−	−	−	+	0.36
		TL-9	+	−	−	+	+	+	+	−	−	−	−	+	−	+	0.50
		VA-11	−	−	−	+	−	−	+	−	−	+	−	+	−	−	0.29
		VI-9	−	−	−	+	−	+	−	−	+	−	−	−	−	+	0.29
		YL-9	+	+	−	−	+	+	+	+	−	−	−	+	+	+	0.64
	Nsp1	QV-9	−	−	−	+	−	+	+	+	−	−	−	+	−	+	0.43
		TV-9	−	−	−	−	−	+	+	−	−	−	+	−	−	+	0.29
		VL-9	+	−	−	+	−	+	−	−	+	−	−	−	−	+	0.36
	Nsp2	FV-9	−	−	−	−	−	−	+	−	−	−	+	+	−	−	0.21
		RT-9	−	−	−	+	−	−	+	−	−	−	−	−	−	−	0.14
		TI-9	+	−	−	−	+	−	+	−	−	−	−	−	−	+	0.29
	Nsp3	FI-10	+	−	+	−	−	+	−	−	−	−	−	−	−	+	0.29
		IV-9	−	−	−	+	+	−	−	−	−	−	−	−	−	+	0.21
		KL-10	−	−	−	+	−	−	−	+	−	−	−	−	−	+	0.21
	Nsp16	QL-9	−	−	−	+	+	+	−	−	−	−	−	+	−	−	0.29
		SL-10	−	−	−	−	+	−	−	−	−	−	−	−	−	−	0.07
		WV-9	−	−	−	−	+	−	−	−	−	−	−	−	−	+	0.14
	33-peptide response rate		0.39	0.06	0.15	0.64	0.58	0.52	0.73	0.27	0.18	0.12	0.12	0.30	0.09	0.73	
	Peptivator S		+	+	−	+	+	+	+	+	+	+	+	+	+	+	0.93
	Peptivator N		+	+	+	+	+	−	+	+	+	−	−	+	−	+	0.71
		CEF	+	+	+	+	+	+	+	+	+	+	+	+	+	+	1.00

#### 3.3.2. Agreement between Algorithms and Immunogenicity

We then sought a correlation by Spearman’s Rho test between the immunogenicity rate and the % Rank assigned by the algorithm of choice for peptide selection (NetMHCpan-4.1b). After removing outlier values from algorithm results, these two parameters showed a significant correlation with a *p* = 0.025, while the comparison between peptide immunogenicity and NetCTLpan MHC % Rank or SYFPEITHI score did not (*p* = 0.081 and *p* = 0.949, respectively), bringing us to elect ex post EL NetMHCpan 4.1b as the best algorithm to predict peptide immunogenicity in our study (Appendix A). Moreover, VaxiJen decision was significantly associated to the immunogenicity rate (*p* = 0.005) (Appendix A).

### 3.4. B Cell Response

Total anti-Spike and anti-NP Abs in plasma samples were tested by commercial assay kits (Figure 3). Antibody titers exhibited a certain variation among subjects, all showing high levels of anti-Spike Abs, far above the cut-off (0.8 U/mL) with a median of 46,140 U/mL (min 964–max 105300). A significant correlation was found between anti-Spike Abs and previous doses of anti-SARS-CoV-2 vaccine (rho = 0.603, *p* = 0.022, by Spearman test, (Appendix A). All subjects also exhibited anti-NP antibody values equal to or above the positivity cut-off (1 U/mL), with a median of 17.48 (min 1, max 286) U/mL.

Furthermore, plasma levels of nAbs was evaluated using a test developed in our laboratory, based on the use of pseudovirus, as described in [73,75]. Anti-Spike nAbs directed towards the Wuhan-1 strain, were analyzed in twelve out of fourteen enrolled subjects, while anti-Spike nAbs against the Omicron BA.1 variant were measured in seven out of fourteen subjects, likely infected with this VOC, as shown in Figure 3. All subjects except one exhibited a high neutralizing titer against the Wuhan-1 strain (median ID50 9789, min 1765, max 41,000). Anti-Wuhan nAbs and anti-Spike Abs strongly correlated (rho = 0.902, *p* < 0.001 by Spearman test, Appendix A).

Similarly, the subgroup of subjects examined for direct Abs against the Omicron BA.1 variant all tested positive (median ID50 3534, min 1450, max 15,762). We also tested the plasma concentration of nAbs directed against other VOCs (Alpha, Delta, Omicron BA.2, and BA.4.5) in a few subjects, selected on the basis of their positive SARS-CoV-2 test date. In all cases, high ID50 values were detected (Appendix A).

### 3.5. Relationship between Symptoms and Demographic, Clinical, and Lifestyle-Related Parameters

Since COVID-19-related symptoms can widely vary within the population, depending on several features, such as individual differences, vaccination status, and infection with different viral variants, we characterized our sample in terms of symptom profiles and their relationship with personal and context-related factors (i.e., age, physical activity, paracetamol intake, vaccine doses, estimated VOC, and number of reported symptoms). To this aim, MCA was applied to the symptom dataset, while personal and context-dependent variables were used as illustrative. The symptomatology pattern among individuals was evaluated in the plane formed by the first two dimensions, which explained 52.25% of the total dataset variability (Dim1 = 30.86%; Dim2 = 21.39%).

As shown in Figure 4, Dim1 can be considered a quantitative indicator of symptomatology (“Size” effect) since it accounts for the differences between individuals who presented many symptoms (severe, positive coordinates) and individuals who did not (mild, negative coordinates); most of the categories marked as ‘Y’ (i.e., presence of symptom) are located in the right plane, with the only exception of the rhinorrhea symptom. Accordingly, Dim1 correlates with the number of reported symptoms (rho = 0.91). Individuals who used paracetamol and practiced sport can be observed at negative values of Dim1, consistently with a picture of mild symptomatology.

Dim2 adds hints on symptoms quality: (1) In the upper-right quadrant, the presence of ‘neurological symptoms’ (i.e., ageusia, anosmia and confusional state) was associated with individuals infected with Delta variant and who received two vaccine doses; (2) in the upper-left quadrant, rhinorrhea was present in a set of individuals infected with the Omicron variant who received a single vaccination dose, (3) in the lower-left quadrant, most of the Omicron-infected subjects who presented the fewest symptoms practiced sport and underwent three vaccine doses; (4) in the lower-right quadrant, individuals presented influenza-like symptoms.

### 3.6. Peripheral Blood T Cell Memory Profile Characterization

Although many studies showed that T cells play an important role in COVID-19 recovery, the contribution of their naïve/memory status is still unclear. We therefore assessed it on both major (total CD3, CD4sp, and CD8sp) and minor (DP1, DP2, DN, and Vδ2) T cells subpopulations, performing an immunophenotyping on fresh whole blood samples of the enrolled subjects, using a seven-color MFC panel (refer to Section 2 Material and Methods, Appendix A) [126,127].

Appendix A describes the distribution of major and minor lymphocyte subpopulations and their corresponding naïve/memory phenotype in our sample (n = 12; 38 total relevant cellular subsets). In particular, scatter plots showing the median and interquartile range of major and minor naïve/memory subsets indicated a prevalence of naïve phenotype within the total CD3+, and CD4+ gated cells, whereas CD8+ T cells showed a higher proportion of TD cells. In contrast, DP1 were mostly represented by CM and EM phenotype, while DP2 were characterized by a higher frequency of EM and TD cells. DN T and γδ T cells presented mostly a TD or EM phenotype.

### 3.7. Host–Pathogen Relationship Described Throughout Linking T Cell Memory Profile to Immune Response and Demographic-Clinical Subject Characteristics

Naïve/memory phenotype results underwent a PCA to investigate the distribution of their pattern, reducing the 38 phenotypic features to 11 Principal Components (PC, Appendix A).

The 53.33% of the total variability in the dataset (scree plot in Appendix A) could be well explained by the first two PCs. In Figure 5A, each phenotypic parameter is represented in a correlation circle by vectors (black arrows), whose coordinates correspond to loadings, i.e., the correlation between the original variable and principal components (PC1, PC2). The figure shows how the memory lymphocytic subpopulations were inter-correlated (the higher the correlation between features, the smaller the angle between arrows) and arranged in the plane according to their maturation status (N, CM, EM, and TD, counterclockwise order).

Subjects represented with a positive pole on PC1 had higher values, as compared to the sample mean, for the variables CD8sp, TD CD3, CD45RA CD3, Lymphocytes, CD3, TD CD8sp, and TD DP1, and lower values for the variables CD4sp, CM DP2, CM CD3, CCR7 CD3, and CM CD8sp (variables selected from the largest loading, as absolute value) (Appendix A). As far the supplementary variables were concerned, the 33-peptide response rate (blue arrow) was correlated to PC1 (rho = 0.51, *p* = 0.09) and associated with the absence of symptom malaise-fatigue (eta2 = 1.74, *p* = 0.085) (Appendix A).

Within PC2, individuals with positive coordinates were characterized by higher values of the variables N CD3, N DP2, N CD8sp, N CD4sp, N DP1, CCR7 CD3, DN, and Vδ2 and lower values for EM CD4sp, CM Vδ2, and DP2 (see Appendix A). This component negatively correlated to age (rho = −0.80, *p* < 0.01; naïve subpopulations were more represented in youngest subjects; green arrow), correlated to the PS-ΔT (rho = 0.54, *p* = 0.054; the largest, the less EM subpopulations; green arrow), and negatively correlated to anti-Spike Ab levels (rho = −0.63, *p* = 0.026; the higher the response, the richer the phenotype in EMs and CMs; red arrow). In our sample, the variable age was correlated to the vax dose# (rho = 0.74, *p* = 0.005) and was negatively correlated to PS-ΔT (rho = −0.7, *p* = 0.010). In summary, the production of anti-Spike antibodies was associated with a phenotype rich in CM and EM (specifically, EM CD3 and CM CD4sp) components and poor in N CD4sp, N DP1, DN, and Vδ2 T cells, along with age, and inversely to PS-ΔT.

In addition, PC2 correlated to the Delta VOC infection (eta2 = 1.929105, *p* = 0.039) and to the confusional state symptom (eta2 = 1.93, *p* = 0.047) (Appendix A). In the PC1-PC2 plane, no categorical variable significantly segregated the individuals.

Further principal components from three through eleven were analyzed to investigate their potential biological/clinical relevance, though each explained a lower percentage of total variance than the first two. Interestingly, PC9 showed maximum correlation with cellular response parameters: a better response to Spike KL-9w, 33-peptides, and, to a lesser extent, to Peptivator N, Spike LA-9, Peptivator S, along with a low response to anti-NP Abs were associated to the positive pole of this dimension (Appendix A), where individuals shared higher values of CM DP2, TD CD8sp, Vδ2, DN, lymphocytes, N DP2, and lower values of N CD8sp and EM DP2 (Appendix A). In addition, PC9 was associated to Omicron VOC infection (eta2 = 0.57, *p* = 0.04 and to Sport_Y (eta2 = 0.58, *p* = 0.06) and the absence of the anosmia symptom (eta2 = 0.64, *p* = 0.04; Appendix A).

When plotting PC2 vs. PC9 dimensions (Figure 5B), it is possible to effectively visualize how the cellular response (blue arrows) to the different viral peptides and proteins was distributed in the same portion of the plane (positive pole of PC9), confirming that subjects tended to respond similarly to all the analyzed cellular stimuli.

The antibody response to the NP protein (dotted red arrow in the negative pole of PC9, Figure 5B), an antigen not included in the vaccine formulation, was clearly opposite to the global cellular response (i.e., the response to Peptivator S and N, KL-9w, LA-9, and the 33-peptide response rate). The immunophenotypic components that seemed to have the greatest weight in directing the cellular response were the Vδ2 and the DN subsets, while N CD8 seemed to be associated with the anti-NP response. Differently, it can also be observed that anti-Spike antibodies and anti-Wuhan nAbs laid very closely in this plane along the negative pole of PC2; in fact, there was a strong correlation between them (*p* <0.001, R2 = 0.860, by Spearman test). As a whole, the antibody response to the Spike protein, which is contained in the vaccine, was related to the age, the vax dose#, and to a shorter PS-ΔT. The lymphocyte subpopulations positively associated with the antibody response to the Spike protein appeared to be EM CD4 and DP2 T lymphocytes, while N CD3, N CD8, and N DP2 were negatively associated. Moreover, in this plane, confusional state during the infection, Omicron/Delta VOC infection and the practice of sport (Figure 5C) clustered individuals into groups (*p* = 0.003, 0.027 and 0.074, by Wilk’s lambda test, respectively).

## 4. Discussion

Although vaccination efforts have been critical in mitigating the spread of the virus and in reducing disease severity, vaccines still need to be updated and redesigned to prevent future infections with emerging mutant strains that may evade the host immune response, as recommended by international health agencies. Vaccination strategies should be conceived to strengthen both the humoral and cellular immune compartments providing a long-lasting protection. In particular, novel T cell-restricted epitopes have proven to be valuable tools for research purposes and promising weapons for vaccines and/or immunotherapeutic interventions against SARS-CoV-2. In this regard, in silico epitope identification is an important step in the vaccine pipeline and it is gaining recognition by both regulatory and funding agencies [128].

The present study has been conducted to predict novel potentially immunogenic viral epitopes by bioinformatics tools and to confirm their immunogenicity on cPBMC derived from a cohort of vaccinated and recovered from mild COVID-19 subjects, as an appropriate population sample to investigate potential immune correlates that are effective in controlling the disease.

To achieve this first aim, we focused on those CD8 T cell HLA-A*02:01 restricted epitopes, poorly studied or not yet tested for their immunogenicity. By in silico prediction, fifteen ancestral Spike peptides, four mutated peptides for *Delta* and/or *Omicron*, and three peptides for each Nsp (Nsp1, Nsp2, Nsp3, and Nsp16) have been identified, with promising prediction scores.

Our results indicated a general low-grade immunogenicity of the selected 9-11mer peptides, except for a peptide namely LA-9 (LLFNKVTLA), starting at position 821 of the S2 Spike protein, that tested positive in 57% of vaccinated and recovered subjects. As far as we know, this is the first report highlighting its remarkable positivity by ex vivo ELISpot assay [97,104].

When thinking about peptide vaccine design, the issue of HLA restriction is a concern, since in the context of this type of vaccine, it would be desirable to consider peptide pools, to cover a certain number of different globally frequent HLA-restricted SARS-CoV-2 immunodominant epitopes.

Using computational algorithms designed to predict the binding affinity with the most common HLA alleles, it was found that LA-9 peptide, beyond its restriction to the HLA-A*02:01 allele, shows at least two other alleles with excellent prediction rankings, one of which is quite prevalent in the population (i.e., HLA-B*08:01). This may explain the ex vivo responsiveness to LA-9 even in some non-HLA-A*02:01 subjects.

Another intriguing issue concerns the immunogenicity of longer protein segments particularly rich in mutations that have arisen from the Wuhan-1 strain to the Delta and Omicron VOCs. Thus, we also evaluated two pairs of Spike LP (one Wuhan-1/Delta and one Wuhan-1/Omicron), derived from two protein regions characterized by a high mutational burden, that, to the best of our knowledge, had not been previously been considered in the literature [129,130]. Although the four selected longer peptides included in our experiments have not been tested in vitro with methodologies designed to amplify the measurable response [129], they showed a low but detectable ex vivo immunogenic capacity by direct cPBMC stimulation. Specifically, the LP 135 derived from the Delta VOC tested positive in 93% of cases, compared to its Wuhan-1 counterpart, which was positive in only 36% of cases. A specific query on the IEDB platform highlighted some peptides, within the longer peptide sequence, with good % Rank predictions for certain alleles highly represented in the population. In fact, some % Ranks were improved by the presence of Delta mutations compared to the Wuhan-1 origin sequence, but none to an extent that would justify the increase in cellular immune response between the two longer peptides. It can be speculated that specific mutations in longer peptides can possibly favor an advantageous architecture for antigen presentation to the immune system, therefore resulting in an enhanced immunogenicity.

The choice of the aforementioned approach, based on a T cell assay, enabled us to characterize the subjects’ responsiveness to the single selected peptides as well as to the overlapping Spike and NP peptide pools. Despite the extensive endeavors of the scientific community, uncertainties persist regarding the relation between disease progression and individual traits, including immune, clinical, and lifestyle factors. The second objective of the study consisted in examining the intricate mutual interconnection among all the collected individual and clinical/immunological features, such as the humoral and cellular response, as well as the peripheral immune profile of naïve/memory T cells.

Overall, the analyzed population sample accurately represents the local epidemiological scenario during the specified period. This snapshot includes individuals who had been vaccinated and subsequently infected, reflecting the real-life conditions when Delta and Omicron VOCs were prevalent. In this context, the immune response to the viral Spike protein resulted from a combination of vaccination and infection responses. However, the immune response to viral proteins other than Spike (e.g., Nsp peptides, Peptivator N, and anti-NP Abs) can solely be attributed to the infection, although it is plausible to hypothesize that its intensity may be modulated by prior vaccination.

Based on these considerations, our results indicated a certain variability across individuals with generally low-grade cellular responsiveness to the majority of the selected 9-11mer and long peptides. Almost all subjects’ cPBMCs responded well to both the Spike and NP peptide pools, with NP eliciting a slightly lower response compared to the Spike protein, as expected [131].

With regard to the humoral response, we observed a strong presence of anti-Spike Abs anti-Wuhan nAbs in almost all patients, which correlated to the number of anti-SARS-CoV-2 vaccine doses received [132], according to the previous literature showing that vaccination against COVID-19 enhances immunity, leading to a significant rise in Spike antibody levels and enhanced neutralizing antibody responses in individuals with a history of mild COVID-19 [133].

The presence and magnitude of antibody responses are known to correlate with various clinical outcomes and disease severity. In particular, during a mild infection, antibody production could be somehow related to the short time frame of infection, to the lower viral load, and to the reduced inflammatory response, as well as to lower antigen exposure compared to the severe course [134]. These factors could also account for the earlier decrease in the titer of anti-NP Abs compared to anti-Spike Abs [135]. Even if a large variability in the anti-NP antibodies has been reported [136], some authors described that low levels of anti-NP Abs are associated with mild course of infection compared to severe clinical presentation [137], in line with our observation. All subjects in our cohort resulted positive for the presence of anti-NP Abs, with variable levels.

High levels of anti-Spike Wuhan nAbs were detected in all subjects included in the study (except the one who had received a single vaccine dose) and well correlated with total anti-Spike Abs, as expected. Overall, the antibody response was lower in younger individuals, who had received fewer vaccine doses, also experiencing a longer time lapse between the infection onset and the sampling date, compared to the elder subjects. Higher antibody titers, particularly of nAbs, have been associated with a more robust immune response and control of viral replication [138]. In fact, nAbs are considered key factors to recovery and protection of the host against SARS-CoV-2, although their long-term response against new variants still remains poorly documented [139].

Regarding the clinical presentation and symptomatology of COVID-19, it is known that it can vary widely among individuals. Common symptoms include fever, cough, shortness of breath, fatigue, and muscle pain. The severity and duration of these symptoms can be influenced by the overall immune response, including both the humoral and cellular components. Rapid virus clearance mediated by SARS-CoV-2-specific T cells prevents severe symptoms of COVID-19 [140]. Moreover, various studies have indicated that symptom profiles may differ between variants of SARS-CoV-2 [141,142,143]. Ultimately, lifestyle factors like diet, exercise, and stress levels can also impact immune function, and thus affect symptomatology [144,145]. Medical history, including pre-existing conditions and prior exposure to related pathogens or vaccines, can also modify susceptibility to and severity of infections [146,147].

A multivariate analysis, focused on symptom profiles of our cohort, showed that most of the overall variability among individuals was associated to the number of symptoms. Furthermore, certain symptoms were more commonly observed in association with a particular viral variant. Specifically, individuals infected with the Delta variant predominantly exhibited ‘neurological’ symptoms, whereas Omicron infection was mostly linked to asymptomatic cases, consistent with previous findings in adult patients [148]. Of note, lifestyle habits, such as intensive sport activity, or the intake of medicaments such as paracetamol were also associated with a very mild symptomatology. However, in a retrospective cohort study of COVID-19 patients, no differences in disease severity were noted between individuals who exclusively used paracetamol and those who did not assume it [149]. Conversely, the consumption of paracetamol was associated with a lower risk of SARS-CoV-2 infection and, in vitro, with a decreased expression of ACE2 protein [150].

Focusing on the analysis of the naïve/memory T cell peripheral asset, our data highlighted that the frequency of major T lymphocyte subpopulations seemed not to substantially differ from that of healthy adult subjects [151,152], even because comparing the observed immune profiles with the existing literature is challenging due to the lack of harmonization in reported results, especially when considering DN and DP minor T cell subsets. A multivariate analysis was then applied to our dataset resembling all the above discussed findings. Individual features such as age, lifestyle, and medical history factors may influence the progression and outcome of a viral infection, and understanding their interconnection might be crucial for developing effective strategies for preventing, diagnosing, and treating infections.

Aging is an extremely sophisticated biological phenomenon that is conditioned by various cellular and systemic alterations, including the suppression of the immune response, even in the context of COVID-19. In fact, age is a risk factor for developing severe COVID-19 outcomes [153], even stronger than vaccination status [154], also due to the potential presence of comorbidities that can affect the immune system [155]. Elderly individuals affected by respiratory diseases generally exhibit a natural decline in immune function over time, with a less robust antibody response and compromised cellular response leading to a higher risk of severe symptoms and complications [156], as well as an increased mortality rate. It is also well-documented that, with an equal vaccination status and an equal time distance from infection, younger subjects demonstrate a superior antibody response [157]. The composition of cell subsets and their memory status across all cell lineages differs between young and elderly individuals [158]. Perhaps the most striking change that occurs within the aging T cell compartment is the decrease in the output of new naïve T cells, as a result of thymic involution [159,160,161] at the end of puberty [162] and at the age of 40–50 [163]. This gradual decline, combined with the accumulation of terminally differentiated memory-like cells in the periphery, with an exhausted and/or senescent profile [164] together with a decrease in proliferation and differentiation of B and T cells in lymph nodes [165] and dysregulation of T cells migration [166] contributes to an overall decrease of the immune response to infections [167,168], limiting the ability to effectively respond to encounters with novel antigens [169]. Memory T cells from older adults generally exhibit diminished proliferative capacity and produce lower levels of cytokines in response to antigenic challenges [170].

In apparent contrast to all these reports, the older subjects in our cohort exhibited a higher expression level of anti-Spike Abs and anti-Wuhan nAbs compared to the younger subjects. They also presented a higher frequency of EM CD4sp and DP2 and a lower proportion of N T cell subsets. One must keep in mind that, in our specific cohort, the younger subjects had been infected earlier in the pandemic, when vaccination had not yet reached a large proportion of the population, and the Delta variant was predominant. The significantly higher number of vaccine doses and the accidentally biasing shorter PS-ΔT of the elderly certainly correlated with the intensity of the humoral response.

In addition, the overall analysis highlighted that the global cellular response was antagonistic to the vaccine-unrelated anti-NP Ab production. Both these factors did not associate with age and the PS-ΔT. The immunophenotypic components that seemed to have influence, to some extent, in directing the global cellular response were the CM DP2, TD CD8sp, Vδ2, and DN subsets. Conversely, subjects more efficient in terms of anti-NP production were characterized by higher frequencies of N CD8 T cells and EM DP2.

In mild cases, a greater proportion of viral antigen-specific CD8 T cells compared with CD4 T cells responses has been described [171]. However, we need to emphasize that our immunophenotypic analysis was based on general T cell populations rather than on antigen-specific T cell subsets.

Following natural infection or vaccination, the generation of effective and persistent T cell memory is essential for long-term protective immunity to the virus. It was proposed that memory T cells might protect populations from severe infections, especially when antibody titers are waning [7,172]. Conflicting results have been reported regarding memory T cell subsets in the context of SARS-CoV-2, probably due to the low number of individuals in the study population and to the lack of grouping by disease severity, age, and sex in some studies. Interestingly, altered percentages of CD4 T cells, CD8 T cells, and their memory subsets were reported to be significantly associated with the disease severity. A clear distinction was observed between memory T cells from individuals with acute severe or acute non-severe COVID-19 and those derived from convalescent and healthy control subjects. In particular, when comparing patients to healthy controls, a significant reduction in the percentage of TD CD8 T cells and an increase in N CD8 T cells were detected. Non-severe patients showed less CD4 and N CD4, and more EM CD4, as well as less CD8, N CD8, and CM CD8 and more TD and EM CD8 T cell frequencies [61]; likewise, the subjects in our cohort that showed a better cellular response, characterized by a higher TD CD8 frequency. However, the correlation between different subsets of memory T cells and COVID-19 severity, and its associated comorbidities, needs further elucidation, especially if considering DP and DN minor T cell subpopulations, which are less described in the published literature. The TD phenotype of subjects showing higher cellular response could be explained with an intense involvement of the CD8 counterpart in contrasting the infection, since it represents the main actor of the cellular response to viruses [173].

Minor T cell subsets can play a significant role in determining the course of a viral infection. Longitudinal studies following patients with primary HIV infection showed an increased frequency of DN T cells [174]. Petitjean et al. demonstrated that the increase in DN T cells producing immunosuppressive cytokines (TGF-β and IL-10) could be involved in the control of harmful immune activation [175]. TCR-γδ+ T cells, which are mostly DN T cells, are potentially related to combating bacterial and viral infections. In cases of infection, such as with the influenza A virus or the *Francisella tularensis* bacterium, they expand rapidly and secrete high amounts of IFN-γ and IL-17A [176]. Our data suggested that peripheral DN and Vδ2 T cells might have a role in the global cellular response, and are inversely correlated with subjects’ age, in agreement with several reports showing their expansion in childhood, and an age-dependent reduction in the periphery [160].

In addition to the already present donor–donor variability, other factors such as age, gender, and body mass index are likely to affect the proportion of DP T cells in the blood. Both DP2 and DP1 T cells were shown to be more prevalent in the blood of healthy older adults compared to young and middle-aged individuals, potentially reflecting long exposures to chronic antigenic stimulation such as cytomegalovirus (CMV). DP T cells were found to be increased in patients with viral infections such as HIV and COVID-19, indicating a potential role of this minor T cell population in the clearance of viruses [54]. Peripheral DP T cells were significantly reduced in severe COVID-19 disease presentations and may be a useful marker to predict disease severity [55]. According to Nascimbeni et al., DP2 are predominantly CM T cells [177]. Indirectly, we could reckon that CM DP2 subset could have a role in favoring the global cellular response in our subjects.

As stated, we found that Omicron infected individuals appeared more prone to develop a virus specific immune response and were associated to a lower number of symptoms, in particular to the absence of the confusion symptom, which can mirror a more severe illness, characterized by neurological compromission. In our cohort, we can assume that the immune response was mainly influenced by the number of vaccine doses, which in turn was mostly associated to elderly and Omicron infected subjects. Consequently, Delta subjects resembled a “younger” T cell phenotype, rich in N, DN, and Vδ2 T cell elements, and showed a certain degree of association with the anti-NP antibodies, antithetically to the cellular and humoral response variables. The success of each VOC compared to the previously dominant is mostly due to altered intrinsic functional properties of the virus and changes in virus antigenicity that allow it to evade a primed immune response [1]. Several studies showed that the overall T cell response induced by infections and first-generation vaccines is preserved against most VOCs, despite the loss of specific responses due to mutations in the immunodominant epitopes that occurs in new variants. Another key reason for the modest impact of variants on T cell immunity is the broad response generated. Each individual mounts responses to 30–40 different epitopes following infection. It appears that antibody evasion and increased transmissibility will continue to be the primary drivers of emerging VOCs rather than significant T cell escape. It is uncertain whether we will observe a gradual and sequential loss of CD8+ T cell epitopes over time, similar to the long-term adaptation seen in H3N2 influenza [178]. There were no significant variations in antibody levels between different breakthrough infection groups based on the vaccine status [179].

Recreational athletes, engaging in regular sports activities for 5–6 h per week, enrolled in our study appeared to exhibit a more favorable cellular immune response, associated with a predominance of peripheral DN and Vδ2 T cells but not with N CD8 T cells, these last appearing to be more associated with young age than with intensive sport practice. Furthermore, they exhibited a reduced number of symptoms during the infection. In general, regular exercise has been related to a reduced risk of moderate COVID-19 severity [180]. Moreover, the severity of COVID-19 in elite athletes, who engage in high-intensity training every day, is predominantly mild and without complications [181] and high-intensity exercise induces strong differential mobilization of CD8 T lymphocyte subsets that exhibit a high effector functionality as well as increased NK levels [182,183,184]. Mild-intensity exercise during COVID-19 improves low-grade systemic inflammation and it is an effective therapeutic strategy to mitigate the severe inflammatory response mediated by SARS-CoV-2 and its consequences, also by modulating Th1/Th2 ratios, often unbalanced in persons at risk of infection and mortality; on the contrary, high-intensity aerobic exercise during the infection may have adverse effects on immune responses [185,186]. All these things considered, we strongly agree with the authors who have emphasized that public health leaders should incorporate physical activity into pandemic control strategies [187].

### 4.1. Pitfalls

The present study exhibits obvious limitations. Despite the statistical significance of some of the data reported herein, the limited sample size has constrained an evident pitfall. To strengthen the statistical validity of the results, it is desirable to include a larger number of participants. Moreover, the study is limited by the lack of basal and follow-up samples as well as by the lack of an uninfected and an unvaccinated subject control group. Only three out of fourteen enrolled subjects were males, making it practically impossible to perform gender-based statistical considerations. Furthermore, the entire set of less-studied peptides with a high affinity prediction for other alleles of the MHC-I system remains unexplored, as we exclusively selected HLA-A*02:01 subjects.

### 4.2. Open Issues and Perspectives

Notwithstanding the numerous published studies on the search for immunogenic epitopes within viral proteins, the huge number of possible epitopes as well as the substantial variability in individual response leave room for the identification of new immunogenic peptides potentially useful for diagnostic and vaccination purposes. In this context, the study of peptide LA-9 could be further explored to ascertain whether it can indeed be considered as a potential additional candidate for a peptide vaccine formulation.

The identification of an immunogenic peptide in a subset of breakthrough COVID-19 cases prompts the exploration of underlying mechanisms and factors contributing to immune response heterogeneity. Factors such as host genetics, prior exposure to related coronaviruses, and variations in immune cell phenotypes may influence the differential immune recognition and response to specific epitopes and deserve more attention to enhance the understanding of immune dynamics and optimize vaccine strategies. Further investigation into the potential of the identified peptide as a vaccine component or as a part of multivalent vaccine formulations is warranted and needed to evaluate the efficacy and safety of peptide-based interventions in clinical settings. Continued exploration of alternative antigens and their potential inclusion in vaccine formulations may enhance overall protection against COVID-19 and related variants.

Other non-structural proteins would have deserved to be tested, such as NSP-6, which has been recently recognized as a key determinant of viral attenuation [188].

Our study did not address the possibility of multiple infections with different SARS-CoV-2 variants within the cohort. Given that participants had a single clinically confirmed infection with a narrow time window between testing positive and negative, multiple variant infections seem unlikely. However, this could be an important consideration for studies with different designs. Investigating multiple infections and their impact on outcomes could offer valuable insights into infection dynamics and vaccine effectiveness [189].

Further investigation is necessary to fully understand the mechanisms underlying the relationships observed between individual characteristics, the immune response to the peptides under study, and their implications in disease progression. This can also help to find vaccination strategies appropriate to the most vulnerable populations. We believe that the determination of the immune profile linked to the memory status of lymphocytes is relevant not only in an infectious context such as COVID-19, but also in other similar scenarios. An in-depth study of immunophenotype and the collection of more life-related information such as physical activity, dietary habits, sleep, hygiene, and so forth, could help increase the effectiveness of immunization strategies. Considering physical activity at various levels of intensity (moderate, intensive, and elite/competitive) could be also of interest [190]. Furthermore, it would be worthy to extend our observations to individuals with different HLA alleles, varying disease stages, presence of comorbidities, and genetic factors. Also, it would be interesting to study the enrolled subjects over time to determine if any cases of long COVID-19 have developed among them and to search for the correlation between the immune response and the persistence of symptoms.

## 5. Conclusions

In conclusion, this paper proposes a novel peptide to be used in the context of peptide vaccine platforms. The identification of new peptides could be important also for monitoring the persistence of the immune response following vaccine-induced immunization and/or infection, for clinical or research purposes. Peptide vaccines may provide long-term cellular immune protection mediated by cytotoxic T cells creating superior resistance to viral mutations, which are currently the greatest threat to the global vaccination campaign. Regarding in silico design of peptide-based vaccines, our opinion is that predictive algorithms are a good selection tool, but they are not sufficient to establish peptide immunogenicity. In fact, our results confirmed the necessity of experimental validation throughout biological assays.

Moreover, a comprehensive understanding of the role of cellular immunity in COVID-19 is crucial for elucidating the pathogenesis of the disease, predicting outcomes, and informing public health interventions. Our results provide valuable insights into the intricate link between alterations in memory T cells and other parameters. To our knowledge, our study can represent a proof-of-concept of the importance of some individual features, such as the peripheral DN/Vδ2 T lymphocyte frequency and the intensive sport activity contribution, to the cellular specific response. Finally, to be prepared for new, undesirable epidemics and pandemics, it is crucial to emphasize the importance of refining studies focused on identifying viral immunodominant peptides and comprehending the individual characteristics that can support effective immune protection against viruses.

## Figures and Tables

**Figure 1 biomolecules-14-01217-f001:**
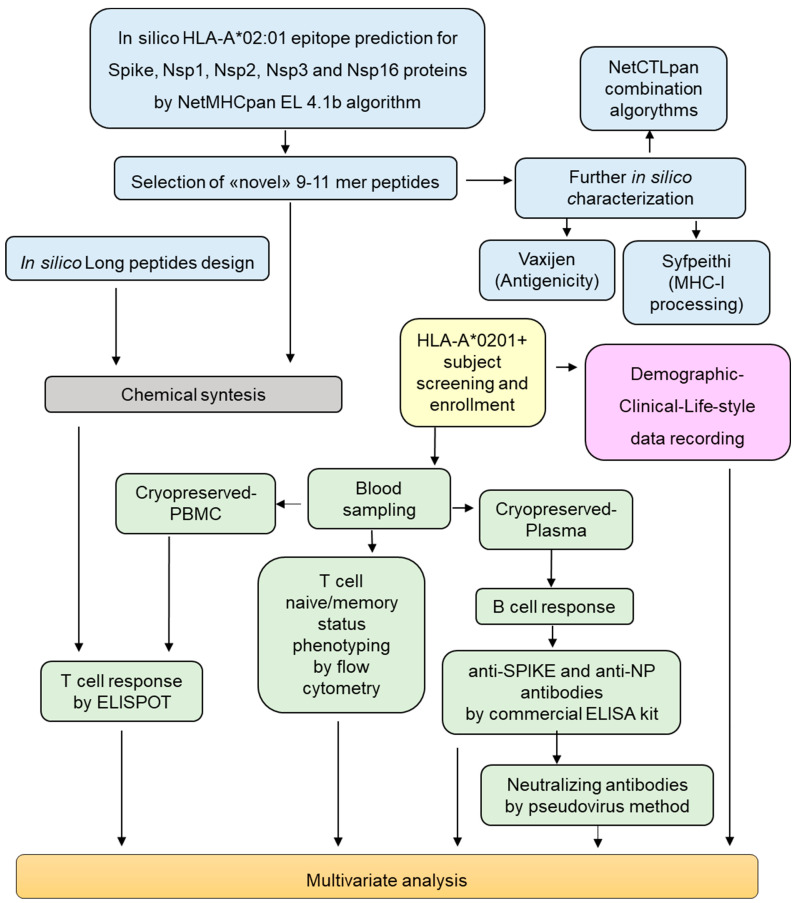
Experimental workflow.

**Figure 3 biomolecules-14-01217-f003:**
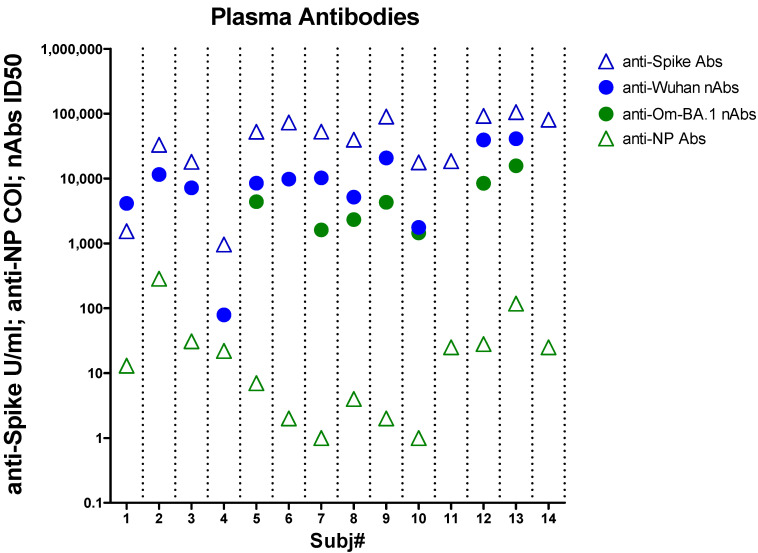
B cell response evaluation. Plasma samples collected 1–6 months after negativization were evaluated for anti-Spike and anti-NP total Ig (IgG, IgM, IgA) antibodies (Abs) by ELISA and for neutralizing antibodies (nAbs) against Wuhan and Omicron BA.1 variant by pseudovirus neutralization assay. Positivity was defined when values were higher than 0.8 U/mL for anti-Spike Abs and 1.0 cut-off index (COI; signal sample/cut-off) for anti-NP Abs. nAb titer is expressed as ID50, corresponding to the dilution of plasma providing 50% inhibition of the infection.

**Figure 4 biomolecules-14-01217-f004:**
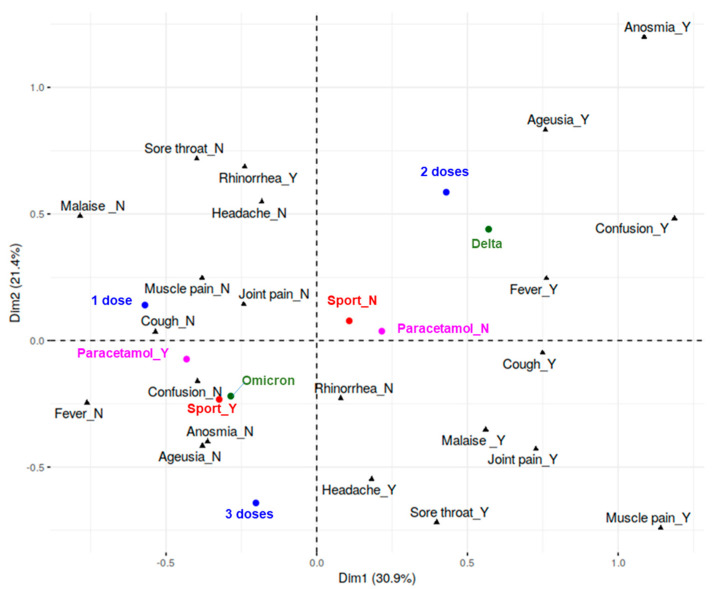
MCA of symptoms. The figure represents the map of categorical variables, both active (symptoms, black) and supplementary (vaccine dose number, blue; VOC, green; intensive sport activity, red; paracetamol intake, pink).

**Figure 5 biomolecules-14-01217-f005:**
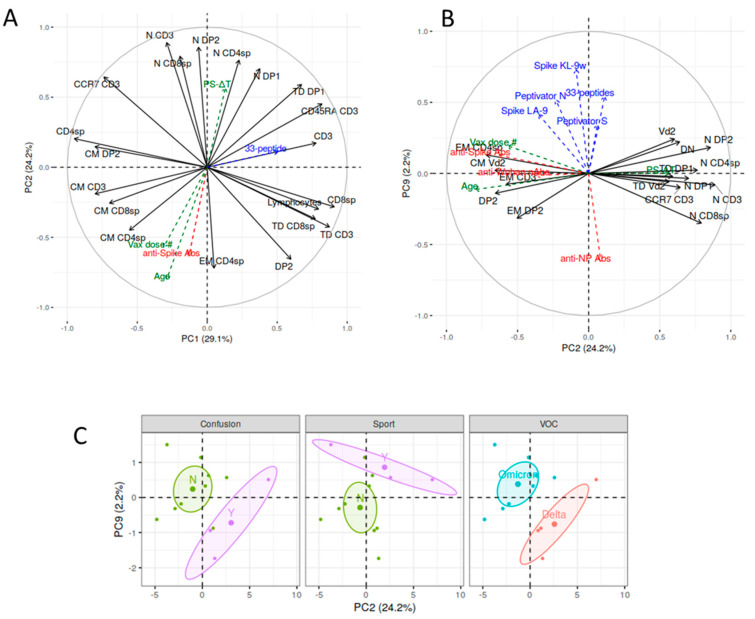
PCA of naïve/memory phenotype and its association with individual and context-dependent parameters. Correlation circle including the 20 most correlated variables (black) to the PCs as well as the ‘response’ and context-dependent quantitative supplementary variables (blue) correlated to PCs; the correlation of the parameters to PCs is represented by the radius; correlation between parameters is represented by the angle between them. (**A**) PC1 vs. PC2 plot. Naïve/memory subset variability as well as antibody response association with phenotype, age, and distance between infection and sampling. Arrow coloring: black, naïve/memory phenotype; blue, T cell response; red, B cell response; green, anagraphic/context related variables. (**B**) PC2 vs. PC9 plot showing cellular response parameter variability (its contrast with anti-NP antibodies) and its association with phenotype. (**C**) Qualitative variables that showed a trend of ability to group subjects in the PC2/PC9 plane.

## Data Availability

The raw data supporting the conclusions of this article will be made available by the authors on request.

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
