# Peer review of "T Cell Peptide Prediction, Immune Response, and Host–Pathogen Relationship in Vaccinated and Recovered from Mild COVID-19 Subjects"

_biomolecules, 2024, doi:10.3390/biom14101217_

Round 1
Reviewer 1 Report
Comments and Suggestions for Authors
The manuscript entitled " T cell peptide prediction, immune response, and host-pathogen relationship in vaccinated and recovered from mild COVID-19 subjects” by Iole et. al, described the identification of several new CD8-restricted immune epitopes of SARS-CoV-2 by using both silico searching method and experimental validation with PBMC samples from recovered patients. Furthermore, they’ve also discussed the demographic differences in immune responses within recovered individuals. The in-silico analysis and experimental data comprehensively support the studies and are of interest to the broad readership of Biomolecules. Thus, I recommend the manuscript be published without further modification.
Author Response
Comment: The manuscript entitled " T cell peptide prediction, immune response, and host-pathogen relationship in vaccinated and recovered from mild COVID-19 subjects” by Iole et. al, described the identification of several new CD8-restricted immune epitopes of SARS-CoV-2 by using both silico searching method and experimental validation with PBMC samples from recovered patients. Furthermore, they’ve also discussed the demographic differences in immune responses within recovered individuals. The in-silico analysis and experimental data comprehensively support the studies and are of interest to the broad readership of Biomolecules. Thus, I recommend the manuscript be published without further modification.
Response: We sincerely thank the reviewer for his/her positive evaluation and for recommending the manuscript publication without further modification. We greatly appreciate the time and effort taken to review our work and we are delighted to hear that our manuscript meets the necessary standards for publication. Thank you again for your feedback and support.
Reviewer 2 Report
Comments and Suggestions for Authors
Overall, I found this to be a comprehensive and well-designed study that was described well.
A couple of specific comments for section 3.2:
1. When listing percentages for vaccine doses and variants, please be consistent. Variants are listed by number of participants while number of vaccination doses and vaccine manufacturer are listed as percentages. The n value for each category is the better choice as proportions from a small cohort can be misleading.
2. Please define PS-deltaT, VS-deltaT, etc. These are not defined in the materials and methods nor in section 3.2 until the table.
A couple of questions I would like the authors to address if appropriate.
1. Did the cohort have access to updated vaccine composition similar to the US and other parts of Europe? If so, which vaccine composition did they receive? Did they receive annual updates?
2. The authors do not address the possibility of multiple infections from different variants. Please discuss this either in the discussion/conclusions or limitations. It is entirely possible there have been multiple infections from multiple variants within the cohort.
Comments on the Quality of English Language
N/A
Author Response
Comment: A couple of specific comments for section 3.2:
- When listing percentages for vaccine doses and variants, please be consistent. Variants are listed by number of participants while number of vaccination doses and vaccine manufacturer are listed as percentages. The n value for each category is the better choice as proportions from a small cohort can be misleading.
Response 1: Thank you for your feedback. We have implemented the requested changes and adjusted Table 2 accordingly. We have used the number of participants for all three variables: vaccine doses, vaccine manufacturers, and variants. Additionally, we have improved Table 2 layout to enhance its readability. Main changes are highlighted in yellow.
- Please define PS-deltaT, VS-deltaT, etc. These are not defined in the materials and methods nor in section 3.2 until the table.
Response 2: Thank you for pointing that out. We have added definitions for PS-DT, VS-DT, and the other acronyms in paragraph 2.7 of the Materials and Methods section, at lines 398-402, highlighted in yellow. This should provide the necessary clarity and consistency throughout the document.
Comment: A couple of questions I would like the authors to address if appropriate.
- Did the cohort have access to updated vaccine composition similar to the US and other parts of Europe? If so, which vaccine composition did they receive? Did they receive annual updates?
Response 1: At the time of the study, all participants received the original formulations of the three vaccine brands. It is very likely that they received updated formulations later on, possibly on an annual basis. However, we cannot confirm this with certainty as the study ended before these updated vaccines became available on the market.
- The authors do not address the possibility of multiple infections from different variants. Please discuss this either in the discussion/conclusions or limitations. It is entirely possible there have been multiple infections from multiple variants within the cohort.
Response 2: Thank you for raising this important point. According to the design of our study, participants had only one clinically confirmed SARS-CoV-2 infection, verified through either molecular or rapid tests, and this infection was localized within a narrow time frame between the initial positive test and subsequent negative test. Based on this design, we believe it is unlikely that participants experienced multiple infections from different variants.
However, we acknowledge that the possibility of multiple infections from different variants is a significant concern and could be highly relevant in studies with a different design. We added a sentence about this topic to the "Discussion/Open Issues and Perspectives" section of the manuscript, specifically in lines 1104-1110, to address this potential issue and its implications, supported by an appropriate literature reference (Diep et al 2024):
Our study did not address the possibility of multiple infections with different SARS-CoV-2 variants within the cohort. Given that participants had a single clinically confirmed infection with a narrow time window between testing positive and negative, multiple variant infections seem unlikely. However, this could be an important consideration for studies with different designs. Investigating multiple infections and their impact on outcomes could offer valuable insights into infection dynamics and vaccine effectiveness (181).
Reviewer 3 Report
Comments and Suggestions for Authors
Review paper Biomolecules
Title of the article: T cell peptide prediction, immune response, and host-pathogen relationship in vaccinated and recovered from mild COVID-19 subjects
Summary
In this article, the authors attempted to identify potentially immunogenic SARS-COV2 peptides in addition to exploring host-pathogen interactions between peripheral immune response, memory profiles, and various demographic, clinical, and lifestyle factors. The authors used both in silico and experimental methods to identify CD8-SARScov2 peptides with unreported immunogenicity. Of these, 15 belonged to the spike protein and three were from NS-proteins (NSP-1,2, 3, and 16). The authors claimed that the peptide LA-9 demonstrated a 57% response rate using an ELISPOT assay, using PBMCs HLA-A*02:01 positive, vaccinated, and mild COVID-19 recovered subjects; hence, the authors indicated that this peptide has potential for use in diagnostic, research, and multi-epitope vaccine platforms.
Major comments:
1. There are other researchers that have found that NSP6 has a role in virulence and that are key determinants of SARS-cov2 attenuation (https://www.nature.com/articles/s41586-023-05697-2). Looking at figure 1, which is much appreciated for understanding the experimental workflow, I see that they have not explored NSP6. It would be good to know why the author has not explored this in this study.
2. In the materials and methods section, the authors mentioned that they used the algorithm NetMHCpan-4.1b They mentioned that the HLA*A0201 allele was acquired and that a list of potential immunogenic peptides for Spike and NSP was obtained. However, it is not clear how they did so. There is no protocol provided to replicate these results. They also mentioned that they used the IEDB website to exclude the most studied peptides, but it is not clear how they explored this website to find such information.
3. The authors also mentioned that they performed mutation analysis of different SARS-cov2 variants of concern using the MEGA software. However, no protocol has been developed to replicate or corroborate these results.
4. Similar to points 2 and 3, there is no protocol or guide for using the following bioinformatics tools: NetCTLpan-1.1, Syfpeithi, and Vaxijen v2.0. The authors mentioned that they use these tools and explain them, but not how they use them.
I would recommend that the authors include these protocols to further review this paper in detail, corroborating the information they claim they have found.
Author Response
Comment 1: There are other researchers that have found that NSP6 has a role in virulence and that are key determinants of SARS-cov2 attenuation (https://www.nature.com/articles/s41586-023-05697-2). Looking at figure 1, which is much appreciated for understanding the experimental workflow, I see that they have not explored NSP6. It would be good to know why the author has not explored this in this study.
Response 1: Thank you for highlighting the potential role of NSP6 in SARS-CoV-2 virulence and for providing the reference to the recent study that identifies it as a key determinant of viral attenuation (https://www.nature.com/articles/s41586-023-05697-2 ).
While we acknowledge the importance of NSP6, we chose not to include it in the current investigation because, at the time of designing the study, we were forced to focus on a limited number of proteins due to budgetary and logistical constraints. Based on our knowledge of the literature then available, the selected non-structural proteins (NSP1-2-3-16) appeared to be the most promising.
That said, we agree that exploring the role of NSP6 in future studies could provide additional insights and contribute to a more comprehensive understanding of SARS-CoV-2 virulence determinants. In this light, we added the following sentence to the “Discussion/Open issues and perspectives” section at lines 1102-1103, highlighted in yellow and supported by the suggested literature reference:
Other non-structural proteins would have deserved to be tested, such as NSP-6, which has been recently recognized as a key determinant of viral attenuation (Chen et al. 2023).
Comment 2: In the materials and methods section, the authors mentioned that they used the algorithm NetMHCpan-4.1b They mentioned that the HLA*A0201 allele was acquired and that a list of potential immunogenic peptides for Spike and NSP was obtained. However, it is not clear how they did so. There is no protocol provided to replicate these results. They also mentioned that they used the IEDB website to exclude the most studied peptides, but it is not clear how they explored this website to find such information.
Response 2: We appreciate the reviewer’s comments regarding the need for greater clarity in the Materials and Methods section, specifically concerning the use of the NetMHCpan-4.1b algorithm and the IEDB website. We understand the importance of providing sufficient detail to ensure our methodology is fully replicable. In response, we added a more detailed description of the process, including the specific parameters used. These changes are located in the Materials and Methods section, specifically on lines 152-186 (highlighted in yellow). We hope that these clarifications sufficiently address the reviewer’s concerns and enhance the reproducibility of our study. The modified lines are as follows:
2.2. Selection and synhtesis of peptides
2.2.1. T cell epitope prediction
To predict potential immunogenic peptides, we used the amino acid sequences of the widely known Spike (YP_009724390.1) and Non-structural protein (Nsp1 - YP_009725297.1, Nsp2 - YP_009725298.1, Nsp3 - YP_009725299.1 and Nsp16 - YP_009725311.1) accession numbers, collected in the database of the National Center for Biotechnological Information (NCBI) and derived from the Wuhan-Hu-1 (genome accession number NC_045512) reference isolate. These sequences were submitted to the NetMHCpan-4.1b EL algorithm available at https://services.healthtech.dtu.dk/services/NetMHCpan-4.1/, which is based on eluted ligands (EL) data and, associating a value to each epitope, it calculates a percentile Rank (% Rank) (66).
The following parameters were set:
- Input type: FASTA
- Allele selected: HLA-A*02:01
- Peptide Length: Any length
- Other fields: default values
The tool generated a list of peptides predicted to bind with high affinity to the HLA-A*02:01 allele, ranked in descending order of % Rank.
2.2.2. Selection of 9-11mer poorly studied ancestral peptides
We then submitted the FASTA format sequence of each listed peptide, with a % Rank below 1.1, to the IEDB Analysis Resource (https://www.iedb.org/result_v3.php?cookie_id=60c6fc&active_tab=Tcell%20Assays) to exclude peptides that had already been shown to elicit a positive response in an IFN-γ ELISPOT release assay by other authors.
The following parameters were set:
- Epitope: Linear peptide
- Sequence: Exact match (inputting the sequence of each peptide)
- Assay: T cell, IFN-γ release ELISPOT
- MHC Restriction: Class I
- Host: Human
- Other fields: Default values
Peptides that had never been reported as positive in IFN-γ-ELISPOT assays (whether tested or not) were selected for synthesis and subsequent immunological testing (by IFN-γ-ELISPOT), as well as for further characterization as described below. The main scientific articles reporting immunogenicity studies of the selected peptides, extracted from the IEDB website after a query without a specified assay type, are also listed in Table 1.
Comment 3: The authors also mentioned that they performed mutation analysis of different SARS-cov2 variants of concern using the MEGA software. However, no protocol has been developed to replicate or corroborate these results.
Response 3: We appreciate the reviewer’s comment regarding the need for additional details about the mutation analysis of different SARS-CoV-2 variants of concern using the MEGA software. We agree that providing a clear protocol is essential for replicating and corroborating our results. We have now added a detailed description of this process, modifying the Materials and Methods section at lines 187-196 (highlighted in yellow). Here we report the modified section:
2.2.3. Selection of 9-11mer mutant peptides
Based on the reference genome accession numbers for Delta B.1.617.2 (MZ359841.1) and Omicron B.1.1.529 (BA.1) (OL672836.1), we selected the mutant Spike protein sequences of the Delta B.1.617.2 (QWK65230.1) and Omicron (BA.1) (UFO69279.1) virus strains and compared them to the ancestral Wuhan-1 strain protein (YP_009724390.1) using the free software Mega 11 (https://www.megasoftware.net/) (67). The software created a new protein sequence alignment by matching the three FASTA format sequences using the ClustalW method. Upon examining the alignment, we identified four mutant peptides corresponding to four selected peptides from the Wuhan-1 strain (KA10w-KA10δ, KL9w-NL9o, VV11w-VV11oδ, VV9w-VG9o).
Comment 4: Similar to points 2 and 3, there is no protocol or guide for using the following bioinformatics tools: NetCTLpan-1.1, Syfpeithi, and Vaxijen v2.0. The authors mentioned that they use these tools and explain them, but not how they use them. I would recommend that the authors include these protocols to further review this paper in detail, corroborating the information they claim they have found.
Response 4: We appreciate the reviewer’s comments regarding the need for additional details on the use of the bioinformatics tools NetCTLpan-1.1, Syfpeithi, and Vaxijen v2.0 in our study. We understand the importance of providing a clear and reproducible methodology for these tools. We have revised and updated the Materials and Methods section on lines 216-243 to include these detailed protocols for using NetCTLpan-1.1, Syfpeithi, and Vaxijen v2.0, ensuring that our methodology is clear and replicable. We hope this additional information addresses the reviewer’s concerns and improves the transparency and reproducibility of our bioinformatics analysis.
Here we report revised lines:
2.2.6. Selected 9-11mer peptide additional characterization
- NetCTLpan version 1.1: Available as a prediction method at the IEDB Analysis resource page (http://tools.iedb.org/netchop/) (67), this tool integrates the prediction of peptide binding affinity to MHC class I molecules within the MHC class I antigen processing pathway. It combines the proteasomal cleavage score (C-score), which predicts the likelihood of protein cleavage at the C-terminus by the proteasome, with the TAP score, which indicates the transport efficiency by the transporter associated with antigen processing (TAP) proteins. For each selected peptide, the sequence in FASTA format was entered into the appropriate input field, the species was set to "human," and the allele was specified as "HLA-A*02:01”, while all other parameters were kept at their default values. The output included a "% Rank", which inversely correlates with the peptide's binding capacity to the MHC molecule of interest.
- SYFPEITHI: This is an online database (syfpeithi.de) that uses an algorithm to assign a score to each amino acid at specific positions based on its frequency in natural ligands, T cell epitopes, or binding peptides (68). Each peptide sequence in FASTA format was submitted to the appropriate input field (Epitope prediction), specifying the HLA-A*02:01 allele and the peptide length. The algorithm produced a score directly proportional to the binding affinity between the MHC molecules and their ligands.
- VaxiJen v2.0 algorithm: Available at https://ddg-pharmfac.net/vaxijen/VaxiJen/VaxiJen.html, this tool evaluates the probability of a given peptide being an antigen based on a trained model (69). Each selected peptide sequence was submitted in the appropriate input field using the default antigenicity threshold setting of 0.4, and the target organism was set to "virus". The output provided an antigenicity score for each peptide along with a qualitative prediction (probable antigen or non-probable antigen).
Please note that, to facilitate any attempts to replicate our results, on line 218, we have replaced the previously indicated NetCTLpan site (https://services.healthtech.dtu.dk/services/NetCTLpan-1.1/ ) with the IEDB site http://tools.iedb.org/netchop/ , as the former has been discontinued. We have successfully verified the identity of the outputs obtained from the two different sites.